# Structural Prognostic Event Modeling for Multimodal Cancer Survival Analysis

**Yilan Zhang[†], Li Nanbo[†], Changchun Yang, Jürgen Schmidhuber, Xin Gao[*]**
[†]Equal contribution, [*]Corresponding author
{yilan.zhang,nanbo.li,changchun.yang,juergen.schmidhuber,
xin.gao}@kaust.edu.sa
King Abdullah University of Science and Technology (KAUST)
Thuwal, Saudi Arabia

## Abstract

The integration of histology images and gene profiles has shown great promise for improving survival prediction in cancer. However, current approaches often struggle to model intra- and inter-modal interactions efficiently and effectively due to the high dimensionality and complexity of the inputs. A major challenge is capturing critical prognostic events that, though few, underlie the complexity of the observed inputs and largely determine patient outcomes. These events—manifested as high-level structural signals such as spatial histologic patterns or pathway co-activations—are typically sparse, patient-specific, and unannotated, making them inherently difficult to uncover. To address this, we propose **Slot-SPE**, a slot-based framework for structural prognostic event modeling. Specifically, inspired by the principle of factorial coding, we compress each patient's multimodal inputs into compact, modality-specific sets of mutually distinctive slots using slot attention. By leveraging these slot representations as encodings for prognostic events, our framework enables both efficient and effective modeling of complex intra- and inter-modal interactions, while also facilitating seamless incorporation of biological priors that enhance prognostic relevance. Extensive experiments on ten cancer benchmarks show that SlotSPE outperforms existing methods in 8 out of 10 cohorts, achieving an overall improvement of 2.9%. It remains robust under missing genomic data and delivers markedly improved interpretability through structured event decomposition. Code available at 
https://github.com/zylvemvet/SlotSPE .

## 1 Introduction

Cancer survival analysis aims to estimate patient mortality risk or survival time, serving as a key tool for personalized prognosis and precision oncology, as it helps quantify outcomes, identify prognostic factors, and guide individualized treatment decisions (Cox & Reid, 2004; Jenkins, 2005; Wiegrebe et al., 2024). Whole-slide images (WSIs) and genomics are two widely-used tools in cancer survival analysis, as they capture complementary prognostic signals: WSIs reveal how cancer manifests morphologically in tissue, while genomics uncovers its underlying molecular drivers. Their integration offers the opportunity for more accurate, robust, and biologically grounded survival prediction, which has in turn sparked growing interest in multimodal learning approaches for survival analysis (Chen et al., 2021b; Xu & Chen, 2023; Jaume et al., 2024; Zhou et al., 2025).

Effective and efficient multimodal learning for survival analysis faces a fundamental challenge: although patient outcomes are often driven by a small number of critical events (Vogelstein et al., 2013; Hosseini et al., 2024), WSI and genomic signals are rather complex and high-dimensional. High-resolution WSIs can exceed $10^5 \times 10^5$ pixels (Song et al., 2023) and genomics involve thousands of genes (Győrffy, 2021). These events manifest as meaningful structural associations within WSI and genomic signals—e.g., in histopathology, immune activity and tumor aggressiveness (Chan et al., 2024; Liang et al., 2025) are reflected in the density and proximity of the pixel organizations of tumor-infiltrating lymphocytes and cancer nests, while in genomics, the tumor-promoting effects emerge from the co-activation of PI3K/AKT and MAPK pathways (Glaviano et al., 2023). How-

ever, learning to uncover such event-related patterns structures from the input signals is difficult for machines: these high-level associations are typically sparse, patient-specific, and unannotated. Compounding the challenge, the enormous scale of input units—millions of WSI pixels and thousands of genomic features—creates massive self-interactions within each modality, further amplified by the need for cross-modal alignment, rendering direct modeling prohibitively expensive.

Existing attempts, e.g., (Ilse et al., 2018; Shao et al., 2021; Chen et al., 2021b; Xu & Chen, 2023; Jaume et al., 2024), fall short of capturing high-level prognostic associations. The most relevant studies are prototype-based methods (Zhang et al., 2024; Song et al., 2024), which employ fixed prototype representations and therefore lack adaptability to individual patients. For example, PIBD (Zhang et al., 2024) approximates a prototypical distribution for each risk group via information bottleneck (Tishby et al., 2000; Alemi et al., 2016). Yet, these prototypes are static-once trained, the same set is applied to all patients. MMP (Song et al., 2024) takes a different route by adopting Gaussian mixture model (GMM) (Bishop & Nasrabadi, 2006) to define prototypes for each WSI. However, the image-wise prototypes are unlearnable (i.e., fixed) during training, making end-to-end optimization suboptimal.

In this paper, we introduce SlotSPE, a slot-based framework for structural prognostic event modeling. Inspired by factorial coding (Schmidhuber, 1992; Higgins et al., 2017; Greff et al., 2020), our method compresses high-dimensional multimodal inputs into a compact set of semantic slots using a slot attention module (Locatello et al., 2020), where each slot corresponds to a latent prognostic event. Unlike fixed prototypes, these slots are dynamically instantiated for each patient, allowing the model to capture individualized prognostic patterns while preserving a concise representation. To reduce slot redundancy and encourage slots to capture distinct prognostic events, we introduce selective slot activation through a Mixture-of-Experts–style decoder (Hampshire & Waibel, 1989; Jacobs et al., 1991; Jordan & Jacobs, 1994; Shazeer et al., 2017), which activates only the most predictive slots for each patient.

Beyond compression, we further model cross-modal interactions by incorporating a biological prior: histopathological phenotypes often reflect underlying molecular events, revealing molecular-morphology mappings (Wang et al., 2021; Liu et al., 2025b). Guided by this insight, we design a cross-modal reconstruction task where omics-derived slots are encouraged to predict gene expression from histology image. This alignment not only enforces biologically meaningful interactions across modalities but also improves robustness in missing genomic data.

The key contributions are: (1) A structural prognostic event modeling framework for multimodal survival prediction that effectively and efficiently extracts patient-specific events; (2) A cross-modal reconstruction that encodes biological priors to enhance alignment and robustness; (3) Extensive evaluation on ten cancer cohorts demonstrating the superiority and robustness; and (4) Enhanced interpretability through the structured decoupling ability of the proposed representation.

## 2 RELATED WORKS

**Single Modality**. The advent of deep learning marks a paradigm shift in cancer prognosis, where artificial neural networks (NN) can uncover prognostic patterns from radiological (Aerts et al., 2014; Lou et al., 2019; Xu et al., 2019; Yao et al., 2021), pathological (Ciresan et al., 2012; Cireşan et al., 2013), and molecular data (Chaudhary et al., 2018; Ching et al., 2018) that were previously inaccessible to conventional methods. In histopathology, whole-slide images (WSIs) under the multiple instance learning (MIL) (Dietterich et al., 1997; Maron & Lozano-Pérez, 1997) paradigm—where a slide is represented as a bag of patches with only slide- or patient-level survival labels—have become standard (Hou et al., 2016; Ilse et al., 2018; Shao et al., 2021; Lu et al., 2021). Recent MIL approaches address the challenge of abundant patches by using clustering (e.g. k-means) (Yao et al., 2020; Vu et al., 2023; Yang et al., 2024; Zhou et al., 2024), graph models (Li et al., 2018; Chen et al., 2021a), and hierarchical WSI representations (Shao et al., 2023). Meanwhile, genomic methods moved from simple MLP or shallow nets on high-dimensional gene profiles (Haykin, 1998; Klambauer et al., 2017) to pathway-based representations that group genes into biologically meaningful sets for better interpretability (Zhao et al., 2021; Hou et al., 2023; Wang et al., 2024). Building on these advances, our method preserves the MIL standard for histopathology, adopts pathway-level genomic features.

**Multiple Modality**. Multimodal approaches, especially the integration of histopathology and genomics, have attracted increasing attention because combining morphological and molecular infor-

mation often yields better prognostic performance than either modality alone (Chen et al., 2020; 2021b; Yuan et al., 2025). Typical strategies include late fusion (e.g., concatenation or Kronecker product) (Chen et al., 2020; Ding et al., 2023; Volinsky-Fremond et al., 2024; Zhao et al., 2025; Luo et al., 2025), and early fusion via co-attention mechanisms operating at the patch-gene level (Chen et al., 2021b; Xu & Chen, 2023; Zhou & Chen, 2023; Jaume et al., 2024; Yuan et al., 2025; Wang et al., 2025; Zhou et al., 2025). Since inputs are complex and high-dimensional, recent work has adopted adaptive token selection (Zhang et al., 2025) or compact latent representations by compressing WSI and pathway data via prototypes (Liu et al., 2025a; Song et al., 2024; Zhang et al., 2024). The former, however, lacks modeling of interpretable structural associations; the latter fails to address sparsity and patient-specific event variation, relying instead on fixed prototypes.

**Structural Event Modeling**. A key challenge in multimodal survival analysis lies in handling the high-dimensional histology and omics inputs, which not only incurs substantial computational costs but also complicates modeling of intricate intra-/inter-modal interactions. An important insight is that patient outcomes are often driven by a small number of critical events (Vogelstein et al., 2013). This motivates approaches that aim to uncover the underlying event-related structures within complex inputs. For examples, PIBD (Zhang et al., 2024) and MMP (Song et al., 2024) attempt to derive structured representations from complex multimodal data. However, relying on fixed prototypes, these methods cannot dynamically capture the sparse and patient-specific structural events that often determine outcomes. From the perspective of machine learning, this challenge can be framed as a problem of factorial coding, a well-established paradigm concerned with decomposing and compressing high-dimensional data into disentangled latent factors (Schmidhuber, 1992; Tipping & Bishop, 1999; Higgins et al., 2017; Kim & Mnih, 2018). Building on this connection, our framework advances beyond static prototypes by introducing learnable slots through slot attention (Locatello et al., 2020), originally developed for object-centric representation learning (Li et al., 2020; 2021; Nanbo et al., 2024). Here, these slots model prognostic events that are selectively activated per patient and aligned via biologically guided cross-modal reconstruction. This design enables a structured decomposition of survival-related events, binding multimodal prognostic signals to interpretable representations.

## 3 METHOD

Survival prediction is the task of estimating a patient risk of an event (e.g., death) given the pathological observations (e.g. WSIs and genomic) of the patient. In this paper, we tackle the problem of learning such hazard predictor from the multi-modal data. Formally, let the hazard prediction function be denoted as $h_t$, representing the probability of death occurring at time t (i.e., survival up to time t). The model is trained using the negative log-likelihood (NLL) loss (Zadeh & Schmid, 2020), denoted as $\mathcal{L}_{\text{surv}}$ (see Appendix B.1 for details). Each patient sample is represented as $\left(\mathbf{X}_h^{(i)}, \mathbf{X}_g^{(i)}, t^{(i)}, c^{(i)}\right)$, where $i$ indexes the patient, $\mathbf{X}_h^{(i)}$ denotes the pathology data, $\mathbf{X}_g^{(i)}$ denotes the genomic profile, $t^{(i)} \in \{1, \ldots, N_t\}$, and $c^{(i)} \in \{0, 1\}$ is the censoring indicator ($c^{(i)} = 0$ for an observed event, $c^{(i)} = 1$ for right-censoring). Operationally, following standard practices in computational pathology (Chen et al., 2021b; Jaume et al., 2024), WSI and genomic inputs are pre-processed by dedicated image and gene encoders and represented as bags of instances: $\mathbf{X}_h^{(i)} = \left\{\mathbf{x}_{h,j}^{(i)} \in \mathbb{R}^d\right\}_{j=1}^{M_h}$ and $\mathbf{X}_g^{(i)} = \left\{\mathbf{x}_{g,j}^{(i)} \in \mathbb{R}^d\right\}_{j=1}^{M_g}$, where $M_h$ and $M_g$ denote the numbers of WSI patches and biological pathways, respectively, and $d$ is the feature dimension. When modeled with state-of-the-art transformers like prior works (Xu & Chen, 2023; Zhou & Chen, 2023; Jaume et al., 2024), the resulting intra- and inter-modal interactions scale quadratically as $O((M_h + M_g)^2)$, creating a major challenge for both efficacy and efficiency.

In this section, we present **SlotSPE**, a novel **Slot**-based **S**turctual **P**rognostic **E**vent modeling framework for multimodal cancer survival analysis (Figure. 1).

### 3.1 SLOT-BASED INFORMATION COMPRESSION

Inspired by factorial coding (Schmidhuber, 1992; Higgins et al., 2017), which aims to disentangle data into independent latent factors, we perform structural decomposition for two bags of histology and genomic instances, i.e., $\mathbf{X}_h^{(i)}$ and $\mathbf{X}_g^{(i)}$, using a slot attention module (Locatello et al., 2020). This allows us to compress the large input sets into much smaller slot-based representations, signif-

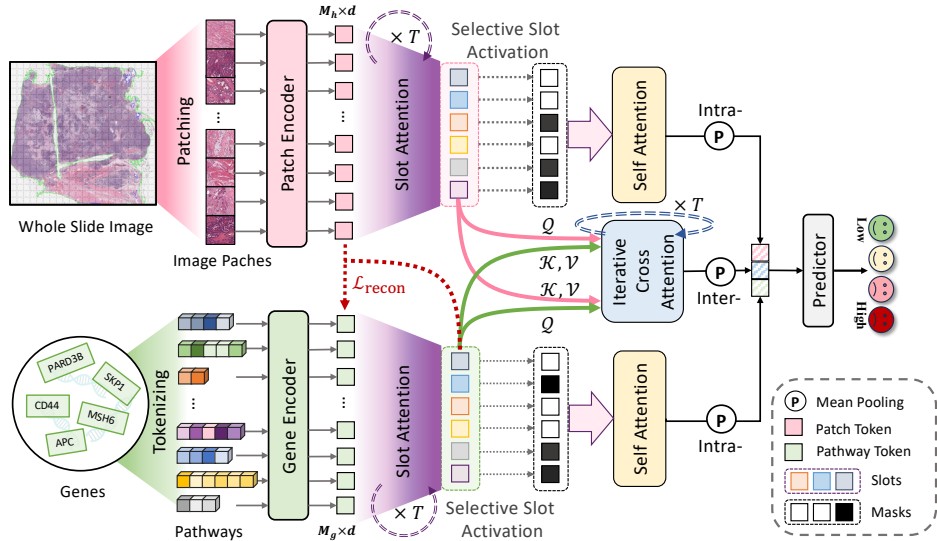

Figure 1: **Framework of SlotSPE**. Histology and gene features are extracted into bag structures, then compressed into slots via slot attention. Selective slot activation enforces sparsity and mutual competition, while a biologically guided cross-modal reconstruction aligns modalities. Finally, slot interactions are modeled using self- and cross-attention to predict survival.

icantly reducing computational overhead. These slots are then decoded using a MoE-style decoder that selectively activates patient-specific slots, promoting both sparsity and competition among them.

**Slot Attention.** Slot attention (Locatello et al., 2020) learns to encode distinct structural patterns present in the input into a finite set of latent vectors, i.e., the "*slots*". Since WSI and genomic data inherently contain massive redundancy (Zhang et al., 2024; Song et al., 2024), it is desirable to aggregate them into a smaller set of compact representations while preserving their semantic content. This makes slot attention particularly well-suited for our task, where the resulting compact representations can be interpreted as latent prognostic events modeled by slots.

We re-organize the input features for both modalities as matrices, $\mathbf{X} \in \mathbb{R}^{M \times d}$. With a set of $S$ slots, organized as $\mathbf{S}^{(0)} \in \mathbb{R}^{S \times d_{\text{slot}}}$, initialized from some learnable distributions, the slot attention compresses $\mathbf{X}$ into $S$ slots through $T$ iterations of cross-attention-based routing. At each iteration $\tau \in \{0, \ldots, T-1\}$, the slots $\mathbf{S}^{(\tau)}$ attend to the input $\mathbf{X}$ to produce updated representations, which are then refined using a multilayer perceptron (MLP). Slot updates across iterations are applied recurrently using a recurrent neural network (RNNs) (Hochreiter & Schmidhuber, 1997; Cho et al., 2014) to preserve temporal consistency and iterative refinement. Mathematically, each slot $\mathbf{s}_k$ is obtained by:

$$\boldsymbol{\alpha}_j^{(\tau)} = \underset{\mathbf{S}^{(\tau)}}{\text{softmax}} \left( \frac{\mathcal{Q}(\mathbf{S}^{(\tau)}) \mathcal{K}(\mathbf{x}_j)}{\sqrt{d_{\text{slot}}}} \right) \in \mathbb{R}^S, \quad \mathbf{u}_k^{(\tau)} = \sum_{j=1}^{M} \alpha_{kj}^{(\tau)} \mathcal{V}(\mathbf{x}_j),$$

$$\mathbf{s}_k^{(\tau+1)} = \text{RNN}\big(\mathbf{s}_k^{(\tau)}, \mathbf{u}_k^{(\tau)}\big), \quad \mathbf{s}_k^{(\tau+1)} \leftarrow \mathbf{s}_k^{(\tau+1)} + \text{MLP}\big(\mathbf{s}_k^{(\tau+1)}\big),$$

(1)

where $\mathcal{Q}(\cdot)$, $\mathcal{K}(\cdot)$, and $\mathcal{V}(\cdot)$ are linear projections into query, key, and value spaces. Note that the softmax is applied over $S$ (slots) for each $j$, enforcing competition among slots and enabling a structured decomposition. Through iterative updates, the learnable bottleneck maps $\mathbb{R}^{M \times d} \to \mathbb{R}^{S \times d_{\text{slot}}}$ with $S \ll M$, yielding $\mathbf{S}^{(T)}$ as the final set of slots (simply as $\mathbf{S}$). The complexity of slot attentions for both histology and genomic instances will be $O(S_g * M_g + S_h * M_h)$.

**Selective Slot Activation**. In our task, to avoid slots redundantly attend to similar structures or collapse onto spurious and prognostic irrelevant patterns, we propose *selective slot activation* (Figure 2(a)): a Mixture-of-Experts (MoE)–style decoder with a learnable gating mechanism that sparsely activates the most predictive slots per patient, encouraging specialization and improving discriminative performance across the cohort.

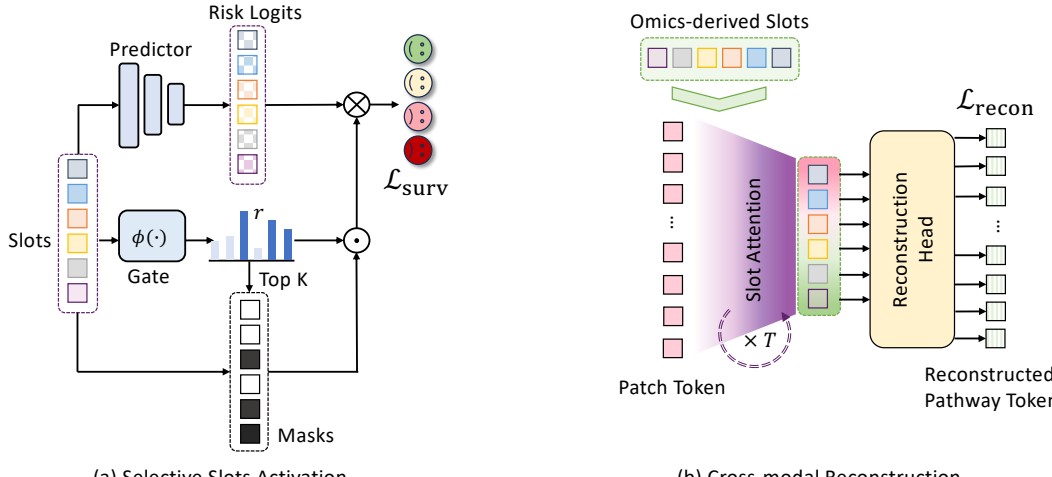

(a) Selective Slots Activation

(b) Cross-modal Reconstruction

Figure 2: Detailed component structure of SlotSPE. (a) Selective slot activation: a Mixture-of-Experts(MoE)–style decoder sparsely activates only the most predictive slots. (b) Cross-modal reconstruction: guided by biological priors, omics-derived slots are aligned with histology by predicting pathway-level gene expression from WSIs.

Conceptually, each slot is treated as an "expert" potentially encoding a critical prognostic event. From the slot set $\mathbf{S}$, we derive per-slot predictions by projecting them into class logits $\boldsymbol{\ell}_k \in \mathbb{R}^{N_t}$ via an MLP predictor, obtaining $S$ candidate predictions. A lightweight gating function $\phi$ predict retention score $r_k$ for each slot $\mathbf{s}_k$:

$$r_k = \phi(\mathbf{s}_k), \qquad \phi : \mathbb{R}^D \to \mathbb{R}. \tag{2}$$

Then the top-$K$ slots are selected using the Gumbel-Top-$K$ trick (Gumbel, 1954; Maddison et al., 2014; Kool et al., 2019) with Straight-Through Estimation (ST) (Jang et al., 2016), producing a differentiable $K$-hot mask $\hat{\mathbf{G}} \in \{0,1\}^S$ (see Appendix B.2 for details). The selected slots are reweighted by masking and renormalizing the softmax weights:

$$w_k = \frac{\tilde{w}_k \cdot \hat{G}_k}{\sum_{s=1}^{S} \tilde{w}_s \cdot \hat{G}_s}, \qquad \tilde{\mathbf{w}} = \mathrm{softmax}(\mathbf{r}) \in \mathbb{R}^S. \tag{3}$$

Finally, patient-level predictions are obtained as a gated mixture of slot logits:

$$y = \sum_{k=1}^{S} w_k \boldsymbol{\ell}_k \in \mathbb{R}^{N_t}. \tag{4}$$

Thus, each patient is represented by a compact, patient-specific subset of slots, yielding a sparse mixture of prognostic events. The entire decoder is trained end-to-end with the survival loss $\mathcal{L}_{\mathrm{surv}}$, which supervises the gating mechanism to prioritize outcome-relevant slots.

**Slots Regularization.** While selective slot activation encourages sparsity, it can also lead to the emergence of null slots—slots that fail to attend to any input features—thereby weakening the competition enforced by the gating mechanism. In information compression, as the entropy of the information bottleneck is upper-bounded by the input features $\mathbf{X}$, non-activated slots—though less predictive than activated ones—are leveraged to capture residual information for input reconstruction, thereby mitigating potential information loss and ensuring completeness. To mitigate this potential loss of information during slot-based compression, we introduce a reconstruction objective. Specifically, we decode the slots using a reconstruction head, and minimize the reconstruction loss $\mathcal{L}_{\mathrm{recon}}(\hat{\mathbf{X}}, \mathbf{X})$, where $\hat{\mathbf{X}}$ denotes the reconstructed features (see Appendix B.3 for details).

### 3.2 MULTIMODAL SLOTS INTERACTION WITH BILOGICAL GUIDANCE

**Cross-modal Reconstruction using Biological Prior**. To ensure effective cross-modal interaction, we introduce cross-modal reconstruction (Figure 2(b)) to uncover molecular-morphology mappings

reflecting the fact that molecular events give rise to histopathological phenotypes (Wang et al., 2021; Liu et al., 2025b).

Given the omics-derived initialization slots $\mathbf{S}_g$, i.e. initialization learned from the training omics data, we perform slot attention (Eq. 1) with histopathology patch features $\mathbf{X}_h$ (for a better interpretability on original WSIs, we choose $\mathbf{X}_h$ instead of using $\mathbf{S}_h$) to obtain cross-modal slot representations $\tilde{\mathbf{S}}_{g \to h}$, which capture gene-related events as expressed in histopathological morphology. Thus, the cross-modal interaction of this step is $O(S_g * M_h)$ with $S_g \ll M_h$. These slots are then decoded by a reconstruction head $\mathcal{R}$ to recover the original genomic features, with position embeddings $\mathbf{Q}_g \in \mathbb{R}^{M_g \times d}$:

$$\mathbf{Q}_g = \text{Embed}([0, \ldots, M_g - 1]). \tag{5}$$

Since the input pathway features are arranged in a fixed index order, corresponds to a specific pathway index, providing a positional prior that anchors reconstruction to the correct genomic coordinate. The reconstructed features are obtained as:

$$\tilde{\mathbf{X}}_g = \mathcal{R}(\mathbf{Q}_g, \tilde{\mathbf{S}}_{g \to h}), \tag{6}$$

with reconstruction loss $\mathcal{L}_{\text{recon}}(\tilde{\mathbf{X}}_g, \mathbf{X}_g)$. Here, histology serves as the morphological manifestation of underlying molecular events, guiding omics-derived slots toward biologically meaningful alignment. Notably, because genomic profiling is substantially more costly than histopathology and often unavailable at test time (Payne et al., 2018), this cross-modal reconstruction enhances robustness: the reconstruction head can impute genomic features directly from histology images, enabling inference even when omics data are missing.

**Slots Interactions.** To capture multimodal dependencies, we model slot interactions at both intra- and inter-modal levels—handled in parallel. Intra-modal interactions are captured using masked self-attention using only the top-$K$ most self-predictive slots within each modality. For inter-modal interactions, we use all slots without masking to fully capture inter-modal synergies, including those arising from slots that are originally less self-predictive within their own modality. Instead of relying on one-shot co-attention (Chen et al., 2021b; Xu & Chen, 2023; Jaume et al., 2024), we design an *iterative cross-attention* scheme that follows the recurrent update in slot attention module. At each iteration, slots from one modality attend to those of the other: histology queries attend to omics keys/values, and vice versa (see Appendix B.4 for details). Finally, the output of three branches are concatenated and passed into a risk classification head for risk prediction. The computational complexity of slots interactions is $O((S_g + S_h)^2)$. Overall, instead of the original $O((M_g + M_h)^2)$, our model requires $O(S_g * M_g + S_h * M_h + S_g * M_h + (S_g + S_h)^2)$. Since $S_g, S_h \ll M_g, M_h$ in practice and the $(S_g + S_h)^2$ term is negligible, this expression can be upper-bounded by $O((S_g + S_h)(M_g + M_h))$.

### 3.3 Training and Inference

**Overall Loss.** The total training objective of SlotSPE is

$$\mathcal{L} = \mathcal{L}_{\text{surv}} + \lambda \mathcal{L}_{\text{recon}}. \tag{7}$$

where $\lambda$ balances the survival and reconstruction components. $\mathcal{L}_{\text{surv}}$ includes both the survival prediction loss and the survival prediction losses w.r.t. each single modalities, while $\mathcal{L}_{\text{recon}}$ combines the slots regularization and cross-modal reconstruction objectives (refer to Appendix B.5 for more details).

**Inference.** At test time, neither slots regularization reconstruction nor cross-modal reconstruction is required when both modalities are available. Cross-modal reconstruction is only invoked when genomic data are missing, serving as an imputation mechanism.

## 4 Experiments

### 4.1 Datasets and Implementations

We perform comprehensive evaluations across ten publicly available cancer cohorts from TCGA[1]. Case counts for each dataset are listed in Table 1. Following (Jaume et al., 2024), we predict disease-

---

[1]https://portal.gdc.cancer.gov/

Table 1: Comparison of mean C-index values across ten cancer datasets. g. and h. refer to genomic modality and histological modality, respectively. The best results and the second-best results are highlighted in **bold** and in underline. (See Table 4 for the completed table with standard deviation.)

| Model | Modality | BRCA (N=1046) | COADREAD (N=573) | KIRC (N=488) | UCEC (N=488) | LUAD (N=467) | LUSC (N=460) | HNSC (N=438) | SKCM (N=409) | BLCA (N=381) | STAD (N=366) | Overall |
|---|---|---|---|---|---|---|---|---|---|---|---|---|
| MLP | g. | 0.644 | 0.661 | 0.750 | 0.731 | 0.634 | 0.584 | 0.584 | 0.652 | 0.677 | 0.615 | 0.653 |
| SNN | g. | 0.645 | 0.664 | 0.754 | 0.753 | 0.652 | 0.548 | 0.597 | 0.673 | 0.694 | 0.585 | 0.656 |
| SNNTrans | g. | 0.657 | 0.710 | 0.747 | 0.724 | 0.648 | 0.555 | 0.605 | 0.684 | 0.699 | 0.593 | 0.662 |
| **SlotSPE** | g. | 0.698 | 0.733 | 0.774 | 0.752 | 0.646 | 0.616 | 0.611 | 0.678 | 0.701 | 0.596 | 0.681 |
| ABMIL | h. | 0.656 | 0.715 | 0.782 | 0.779 | 0.645 | 0.593 | **0.664** | 0.668 | 0.663 | 0.642 | 0.681 |
| TransMIL | h. | 0.647 | 0.723 | 0.767 | 0.759 | 0.655 | 0.582 | 0.644 | 0.632 | 0.647 | 0.656 | 0.671 |
| CLAM-SB | h. | 0.696 | 0.742 | 0.784 | 0.757 | 0.639 | 0.584 | 0.649 | 0.643 | 0.645 | 0.637 | 0.678 |
| CLAM-MB | h. | 0.715 | 0.721 | 0.782 | 0.788 | 0.632 | 0.608 | 0.636 | 0.640 | 0.657 | 0.641 | 0.682 |
| **SlotSPE** | h. | 0.738 | 0.729 | 0.790 | 0.779 | 0.656 | 0.598 | 0.635 | 0.659 | 0.642 | **0.676** | 0.690 |
| Porpoise | g.+h. | 0.651 | 0.699 | 0.788 | 0.788 | 0.647 | 0.597 | 0.599 | 0.682 | 0.690 | 0.619 | 0.676 |
| MCAT | g.+h. | 0.709 | 0.737 | 0.795 | 0.802 | 0.659 | 0.583 | 0.622 | 0.667 | 0.688 | 0.639 | 0.690 |
| MOTCat | g.+h. | 0.717 | 0.742 | 0.799 | 0.773 | 0.663 | 0.573 | 0.627 | **0.688** | 0.700 | 0.642 | 0.692 |
| CMTA | g.+h. | 0.731 | 0.720 | 0.794 | 0.764 | 0.662 | 0.599 | 0.619 | 0.681 | 0.702 | 0.639 | 0.691 |
| SurvPath | g.+h. | 0.728 | 0.727 | 0.787 | 0.751 | 0.661 | 0.577 | 0.611 | 0.666 | 0.666 | 0.647 | 0.682 |
| PIBD | g.+h. | 0.662 | 0.716 | 0.772 | 0.783 | 0.634 | 0.575 | 0.627 | 0.649 | 0.693 | 0.644 | 0.676 |
| MMP | g.+h. | 0.714 | 0.686 | 0.770 | 0.755 | 0.634 | 0.580 | 0.580 | 0.682 | 0.686 | 0.602 | 0.669 |
| LD-CVAE | g.+h. | 0.705 | 0.753 | 0.792 | 0.788 | 0.651 | 0.620 | 0.635 | 0.682 | 0.635 | 0.653 | 0.692 |
| **SlotSPE** | g.+h. | **0.779** | **0.773** | **0.815** | **0.813** | **0.683** | **0.634** | 0.642 | **0.688** | **0.708** | 0.671 | **0.721** |

specific survival (DSS) with histological data including all diagnostic WSIs and transcriptomic profiles with DSS labels are obtained from cBioPortal[2]. Pathway gene sets are curated from Hallmarks (Subramanian et al., 2005; Liberzon et al., 2015) and Reactome (Gillespie et al., 2022), with genes absent in cBioPortal removed. We use 5-fold cross-validation and report the concordance index (C-index) (Harrell Jr et al., 1996) with standard deviation (std) and visualize the Kaplan–Meier (KM) Kaplan & Meier (1958) survival curves for stratified risk groups, perform the log-rank test Mantel et al. (1966) and restricted mean survival time (RMST) (Irwin, 1949; Karrison, 1986) to assess differences between them. Further information can be found in Appendix C.

## 4.2 COMPARISONS WITH SOTAS

We benchmark our approach against three categories of state-of-the-art (SOTA) methods: (1) **Transcriptomics**. We consider a feed-forward neural network (MLP) (Haykin, 1998), a pathway-specific baseline (SNNs) (Klambauer et al., 2017), and SNNTrans, which integrates SNNs with transformers (Klambauer et al., 2017; Shao et al., 2021). (2) **Histology**. We compare with leading MIL-based models, including ABMIL (Ilse et al., 2018), TransMIL (Shao et al., 2021), and CLAM (Lu et al., 2021). (3) **Multimodal**. We evaluate against eight multimodal SOTA methods including Porpoise (Chen et al., 2022), MCAT (Chen et al., 2021b), MOTCat (Xu & Chen, 2023), CMTA (Zhou & Chen, 2023), SurvPath (Jaume et al., 2024), PIBD (Zhang et al., 2024), MMP Song et al. (2024), and LD-CVAE (Zhou et al., 2025). Among them, the first five methods adopt instance-level interactions for histology and omics integration, PIBD and MMP employ prototype-based modeling, while LD-CVAE is specifically designed to handle missing genomic data.

From Table 1, we observe that most multimodal approaches (except Porpoise, PIBD, and MMP) outperform unimodal ones, confirming the complementary value of integrating histology and genomics. Although prototype-based methods (PIBD and MMP) use prototypes to represent complex inputs, their limited adaptability in capturing patient-specific events leads to unsatisfactory results on certain datasets, reducing overall effectiveness. In contrast, **SlotSPE** consistently achieves superior results, ranking first or second in nearly all cohorts and outperforming the second-best method by 2.9% in overall C-index across ten cancer types. Notably, this advantage also holds in *single-modality* settings: gene-only variant surpasses the next best model by 1.9%, while WSI-only variant improves by 0.8% in overall C-index. These results highlight the generalizability and flexibility of SlotSPE on cancer types and modalities. By dynamically capturing structured prognostic events and enabling cross-modal event-level alignment, SlotSPE establishes itself as a strong and reliable framework for survival prediction.

---

[2]https://www.cbioportal.org/

Table 2: C-index across ten cancer datasets under missing-genomics settings. The column *Missing* indicates whether the genomic is missing (✓ = missing). (See Table 6 for the complete table.)

| Model | Missing | BRCA | COADREAD | KIRC | UCEC | LUAD | LUSC | HNSC | SKCM | BLCA | STAD | Overall |
|---|---|---|---|---|---|---|---|---|---|---|---|---|
| LD-CVAE | | 0.705 | 0.753 | 0.792 | 0.788 | 0.651 | 0.620 | 0.635 | 0.682 | 0.635 | 0.653 | 0.692 |
| LD-CVAE | ✓ | 0.715 | 0.736 | 0.798 | 0.775 | 0.638 | 0.596 | **0.649** | 0.676 | 0.644 | 0.647 | 0.688 |
| SlotSPE | | **0.779** | **0.773** | **0.815** | **0.813** | **0.683** | **0.634** | 0.642 | **0.688** | **0.708** | 0.671 | **0.721** |
| SlotSPE | ✓ | 0.734 | 0.741 | 0.789 | 0.802 | 0.663 | **0.634** | **0.649** | 0.678 | 0.678 | **0.677** | 0.704 |

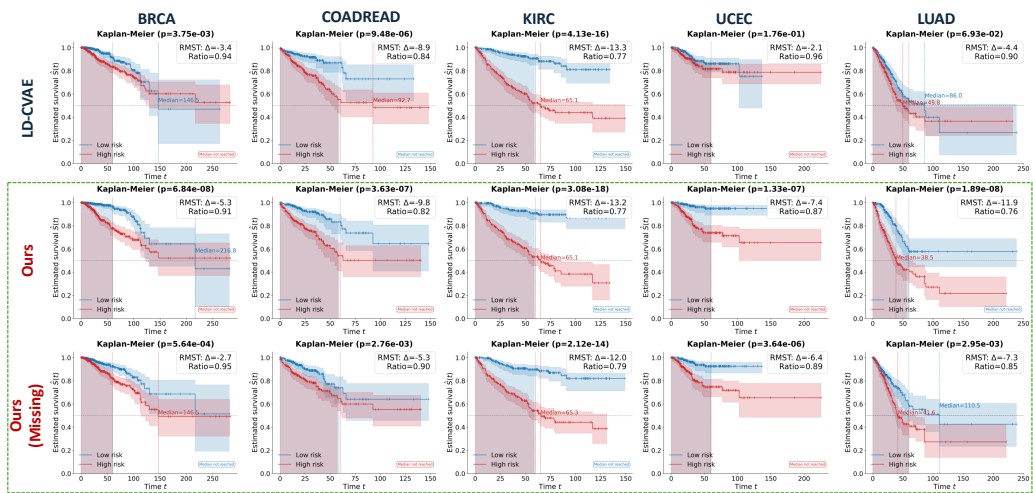

Figure 3: Kaplan–Meier curves of predicted high-risk and low-risk groups. A $p$-value $< 0.05$ at the top indicates statistically significant separation between groups. The restricted mean survival time (RMST) up to 60 months is also reported, with values shown as $\Delta$ (High–Low), and ratio (High/Low). (Zoom in to view details.)

## 4.3 ROBUSTNESS

Since collecting genomic data is substantially more costly than acquiring histology images (Payne et al., 2018), clinical deployment often faces performance degradation when genomic features are unavailable. To evaluate the robustness of our approach, we test SlotSPE under the missing-genomic-modality setting (Table 2). SlotSPE maintains strong performance, and in several cohorts (e.g., HNSC, STAD) its missing-modality variant matches or even surpasses the full-modality version, demonstrating resilience to incomplete omics data. This robustness stems from the cross-modal reconstruction task, which enables the model to infer pathway representations from histology through omics-derived slots. Compared with LD-CVAE (Zhou et al., 2025)—a method explicitly designed for missing genomic data—SlotSPE shows a somewhat larger drop relative to its full-modality variant, yet still achieves superior or comparable performance across most datasets. Notably, its overall performance even exceeds that of full-modality LD-CVAE. These findings highlight that SlotSPE not only integrates multimodal features effectively but also delivers robust predictions in the absence of genomic data.

## 4.4 RISK STRATIFICATION

We further evaluate our model using standard survival analysis techniques. Figure 3 shows Kaplan–Meier (KM) curves, where patients are stratified into high- and low-risk groups based on predicted risk scores, with validation-set median as cut-off. Group separation is assessed using the log-rank test, and the corresponding $p$-values are reported. Across cohorts, our method achieves significantly greater separation between risk groups than the second-best method (LD-CVAE). For instance, in UCEC, LD-CVAE yields a non-significant $p$-value ($> 0.05$), while our model demonstrates significant separation ($p = 1.33 \times 10^{-7}$). For a more intuitive quantitative view, our method attains a larger absolute RMST difference $\Delta$ (High–Low) and a smaller RMST ratio (High/Low), together indicating robust prognostic stratification—even under missing-modality condition.

Table 3: Ablation of model components reported as mean C-index across ten cancer datasets. (See Table 8 for the complete table.)

| Variants | BRCA | COADREAD | KIRC | UCEC | LUAD | LUSC | HNSC | SKCM | BLCA | STAD | Overall |
|---|---|---|---|---|---|---|---|---|---|---|---|
| Baseline (Vanilla Slots) | 0.711 | 0.734 | 0.784 | 0.779 | 0.661 | 0.592 | 0.610 | 0.668 | 0.686 | 0.644 | 0.687 |
| w/o Selective Slot Attention | 0.728 | 0.755 | 0.787 | 0.800 | 0.659 | 0.620 | 0.615 | 0.685 | 0.688 | 0.650 | 0.699 |
| w/o Slots Regularization | 0.734 | 0.763 | 0.794 | 0.811 | 0.662 | 0.611 | 0.621 | 0.679 | 0.693 | 0.667 | 0.704 |
| w/o Cross-modal Reconstruction | 0.710 | 0.758 | 0.787 | 0.794 | 0.663 | 0.619 | 0.611 | 0.671 | 0.698 | 0.645 | 0.696 |
| w/o Iterative Cross-attention | 0.727 | 0.761 | 0.803 | 0.794 | 0.653 | 0.602 | 0.624 | 0.676 | 0.702 | 0.648 | 0.699 |
| SlotSPE | **0.779** | **0.773** | **0.815** | **0.813** | **0.683** | **0.634** | **0.642** | **0.688** | **0.708** | **0.671** | **0.721** |

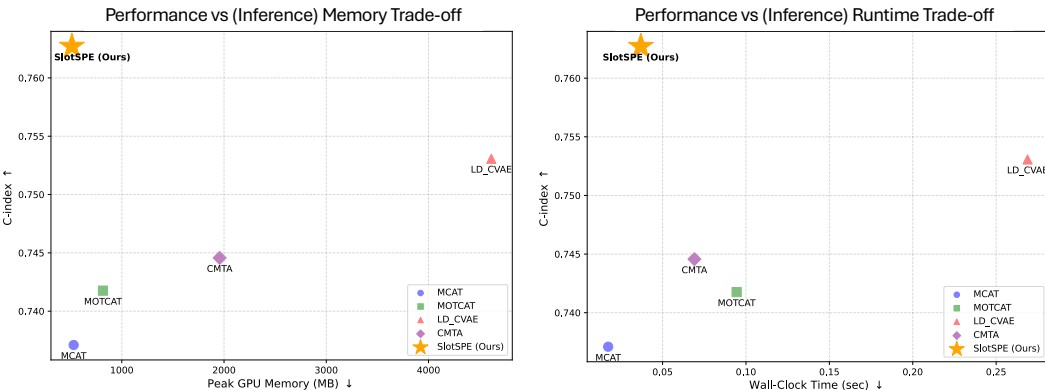

Figure 4: Performance vs (Inference) Memory/Runtime Trade-off

## 4.5 ABLATION STUDY

Table 3 summarizes the ablation results for each component of SlotSPE. We construct a strong baseline by retaining the vanilla slot attention and multimodal fusion modules while removing *selective slot attention* and *reconstructions*, and replacing the *iterative cross-attention* in multimodal fusion with standard dual branch cross-attention. As shown in the first and the last rows, SlotSPE yields a substantial improvement over this baseline. Moreover, removing any of the four key components of SlotSPE results in a consistent performance drop across most datasets (Rows 2–6), indicating that each module contributes critically to the framework. Specifically, excluding selective slot attention reduces the model's ability to focus on discriminative prognostic patterns; removing cross-modal reconstruction with biological priors weakens the alignment between histology and genomics; and dropping iterative cross-attention impairs the iterative refinement needed to capture complex cross-modal dependencies. Slots regularization reconstruction further helps prevent null slots. Overall, the full SlotSPE achieves the best performance, with an average C-index of 0.721, demonstrating that all four modules act synergistically to enable effective multimodal survival prediction.

## 4.6 EFFICIENCY

In this section, we assess the inference efficiency of our model by comparing it against the four strongest baselines listed in Table 1. Because all methods share the same WSI and omics encoders, we exclude these components and evaluate only the model-specific modules to ensure a fair and meaningful comparison. We measure peak inference memory and wall-clock time on the COAD-READ dataset, using all WSI patches during inference and configuring SlotSPE with 16 slots and 3 iterations. As shown in Figure 4, SlotSPE achieves the highest C-index while maintaining low runtime and memory usage. MCAT and MOTCAT achieve shorter runtime primarily because they omit self-attention computations within WSI patches; however, this design choice leads to a considerable drop in predictive performance. CMTA improves upon these baselines by employing Nyström attention (Xiong et al., 2021) as an approximation to full self-attention, but its capacity remains insufficient to capture the sparse, patient-specific structural prognostic signals. LD-CVAE attains the second-best performance, yet its reliance on a variational autoencoder introduces substantial memory and runtime overhead. Overall, SlotSPE offers the most favorable balance between efficiency and predictive performance, with minimal memory usage and the second-fastest runtime among all compared methods. Additional analysis is provided in the Appendix G.

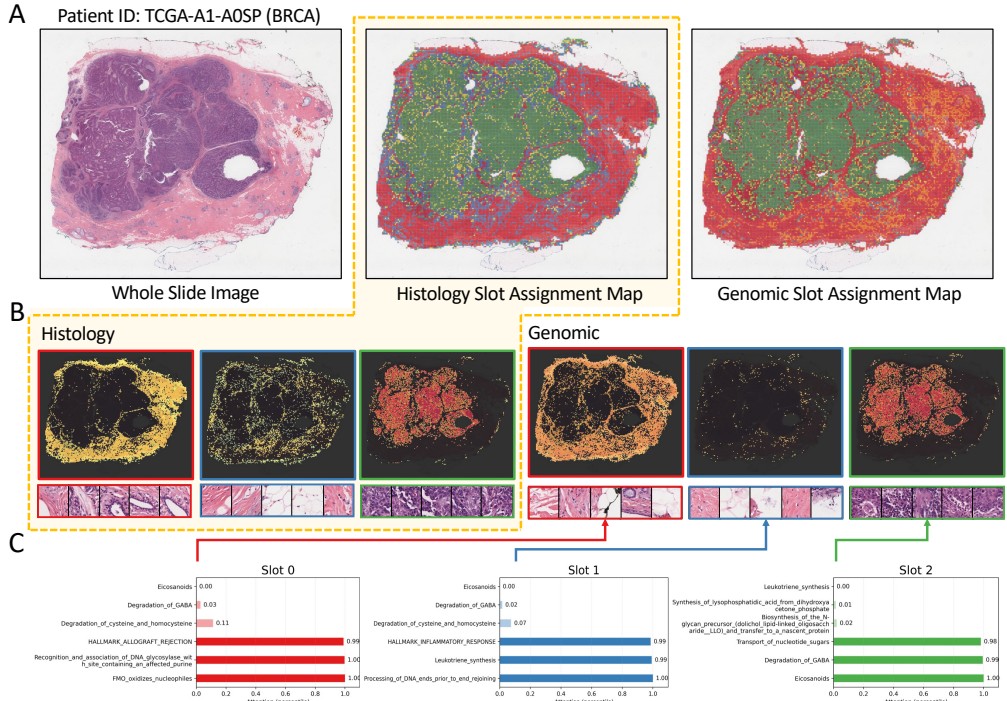

Figure 5: Interpretability of slots. (A) Original WSI, assignment map of histology-derived slots, and assignment map of omics-derived slots. (B) Attention maps for each slot with the top-5 most relevant patches highlighted. (C) Top-3 relevant and irrelevant pathways identified per slot.

## 4.7 INTERPRETABILITY

**Cross-modality Alignment.** Visualization of slot assignments (Figure 5A) shows that histology- and genomics-derived slots often align to structurally similar tissue regions in WSIs, reflecting event-level cross-modal consistency. Subtle differences remain, which can be attributed to modality-specific information: morphology-driven slots emphasize structural patterns, whereas genomics-driven slots highlight regions linked to molecular features. **Intra- and Inter-modal Interactions.** For intra-modal analysis, we visualize attention maps of histology-derived slots (first three plots in Figure 5B) and report the top-3 most relevant and irrelevant pathways per omics-derived slot (Figure 5C). For inter-modal analysis, we link pathway-level attention scores with the corresponding genomic-slot attention maps on WSIs (last three plots in Figure 5B). For instance, in Slot 2, highly attended pathways correspond to tumor-enriched regions, consistent with prior studies on breast cancer prognosis (Wang & DuBois, 2010; Neman et al., 2014; Chen et al., 2019; Ta et al., 2021; Xie et al., 2023). In general, these interpretability analyzes demonstrate the ability of SlotSPE to establish event-level alignment across modalities and reveal pathway–morphology correspondences, offering the potential for biologically plausible explanations and new hypotheses for further validation. Additional examples are provided in the Appendix I.

## 5 CONCLUSION

In this work, we introduced **SlotSPE**, a slot-based framework for structural prognostic event modeling in multimodal cancer survival analysis. By compressing high-dimensional histology and omics data into interpretable slots, selectively activating the most predictive ones, and aligning them through biologically guided cross-modal reconstruction, SlotSPE effectively captures sparse, patient-specific prognostic events and effectively reduces the computational complexity of multimodal interactions. Our extensive evaluation across ten TCGA cohorts demonstrates that SlotSPE not only achieves state-of-the-art predictive performance but also maintains robustness under missing-modality conditions. Moreover, the interpretability analyses highlight its ability to uncover event-level morphology–molecular correspondences. These results underscore the potential of SlotSPE as a powerful and interpretable framework for multimodal precision oncology.

ACKNOWLEDGMENT

We gratefully acknowledge Dr. Juan He from the Department of Pathology, the First Affiliated Hospital of Guangxi Medical University, for her assistance in providing the patch annotations for our visualizations. This work was supported by the King Abdullah University of Science and Technology (KAUST) Office of Research Administration (ORA) under Award Nos. REI/1/5234-01-01, REI/1/5414-01-01, REI/1/5289-01-01, REI/1/5404-01-01, REI/1/5992-01-01, and URF/1/4663-01-01. Additional support was provided by the Center of Excellence for Smart Health (KCSH) under Award No. 5932, and the Center of Excellence on Generative AI under Award No. 5940.

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

APPENDIX TABLE OF CONTENTS

## A  THE USE OF LARGE LANGUAGE MODELS (LLMS)

In this work, we use Large Language Models (LLMs) to correct grammar mistakes and polish the writing.

## B  METHOD DETAILS

### B.1  PROBLEM FORMULATION

Given the $i$-th patient with multimodal inputs $\mathbf{I}^{(i)} = \left(\mathbf{X}_h^{(i)}, \mathbf{X}_g^{(i)}, t^{(i)}, c^{(i)}\right)$ where $\mathbf{X}_h^{(i)}$ denotes the pathology data, $\mathbf{X}_g^{(i)}$ denotes the genomic profile, $t^{(i)} \in \{1, \ldots, N_t\}$ is the discretized survival time interval (risk band), and $c^{(i)} \in \{0, 1\}$ is the censoring indicator ($c^{(i)} = 0$ for an observed event, $c^{(i)} = 1$ for right-censoring). Following the multiple instance learning (MIL) paradigm (Chen et al., 2021b; Jaume et al., 2024), features extracted from the WSI patch encoder and gene encoder are organized as bags of instances: $\mathbf{X}_h^{(i)} = \left\{\mathbf{x}_{h,j}^{(i)} \in \mathbb{R}^d\right\}_{j=1}^{M_h}$ and $\mathbf{X}_g^{(i)} = \left\{\mathbf{x}_{g,j}^{(i)} \in \mathbb{R}^d\right\}_{j=1}^{M_g}$, where $M_h$ and $M_g$ denote the numbers of WSI patches and biological pathways, respectively, and $d$ is the feature dimension. The objective is to learn $\mathcal{F}_\theta : \mathcal{X}_h \times \mathcal{X}_g \to \mathbb{R}^D$ mapping $(\mathbf{X}_h^{(i)}, \mathbf{X}_g^{(i)}) \mapsto \mathbf{z}^{(i)}$ for survival prediction. We denote by $h_t^{(i)}$ the predicted hazard probability at a discrete time interval $t$, i.e., the probability of an event occurring at $t$ given survival up to $t$. The survival function is $\mathcal{S}_t^{(i)}$. With $N_D$ patients in the training set, the hazard, survival, and the negative log-likelihood (NLL) loss (Zadeh & Schmid, 2020) are:

$$h_t^{(i)} = P(T = t \mid T \geq t, \mathbf{z}^{(i)}), \quad \mathcal{S}_t^{(i)} = \prod_{k=1}^{t} \left(1 - h_k^{(i)}\right),$$

$$\mathcal{L}_{\mathrm{surv}}(\{\mathbf{z}^{(i)}, t^{(i)}, c^{(i)}\}_{i=1}^{N_D}) = -\sum_{i=1}^{N_D}\Big[c^{(i)}\log\mathcal{S}_{t^{(i)}}^{(i)} + (1 - c^{(i)})\log\mathcal{S}_{t^{(i)}-1}^{(i)} + (1 - c^{(i)})\log h_{t^{(i)}}^{(i)}\Big].$$

$$(8)$$

### B.2  DETAILS OF SELECTIVE SLOT ACTIVATION

To achieve sparse yet differentiable slot selection, we adopt the Gumbel-Top-$K$ (Gumbel, 1954; Maddison et al., 2014; Kool et al., 2019) trick combined with a Straight-Through (ST) estimator (Jang et al., 2016).

Given slot scores $\mathbf{r} \in \mathbb{R}^S$, we add i.i.d. Gumbel noise $g_k \sim \mathrm{Gumbel}(0, 1)$:

$$\tilde{r}_k = r_k + g_k. \tag{9}$$

This perturbation transforms deterministic scores into a stochastic sampling process where larger $r_k$ values are more likely to be selected. To obtain a differentiable relaxation, we then apply softmax to the perturbed scores:

$$\boldsymbol{\alpha} = \mathrm{softmax}(\tilde{\mathbf{r}}), \tag{10}$$

In practice, we use an ST estimator that combines a hard mask for the forward pass with a soft relaxation for the backward pass:

$$\hat{\mathbf{G}} \leftarrow \mathrm{onehot\_topk}(\boldsymbol{\alpha}) + \left(\boldsymbol{\alpha} - \mathrm{stopgrad}(\boldsymbol{\alpha})\right). \tag{11}$$

where $\mathrm{onehot\_topk}(\cdot)$ produces a discrete $K$-hot mask in the forward pass, while gradients flow through the soft assignment $\boldsymbol{\alpha}$.

Finally, the selected slots are reweighted by masking and renormalizing the softmax weights in Eq. 3, which ensures that only the top-$K$ slots contribute, while their relative importance is adaptively normalized. The resulting $\hat{\mathbf{G}} \in [0, 1]^S$ acts as a differentiable $K$-hot mask, enforcing sparse slot selection while preserving end-to-end trainability.

### B.3 DETAILS OF RECONSTRUCTIONS

**Slots Regularization as Reconstruction**. Within each modality, slots $\mathbf{S}$ are required to reconstruct the original modality-specific features to address the emergence of null slots. We use a Transformer decoder (Vaswani et al., 2017) as the reconstruction head. For genomics, slot embeddings $\mathbf{S}_g$ are decoded to approximate the original pathway embeddings $\mathbf{X}_g$, guided by the positional embeddings $\mathbf{Q}_g$ (Eq. 5), with reconstruction optimized via MSE loss:

$$\hat{\mathbf{X}}_g = \mathcal{R}(\mathbf{Q}_g, \mathbf{S}_g), \qquad \mathcal{L}_{\text{recon}}^g = \|\hat{\mathbf{X}}_g - \mathbf{X}_g\|_2^2. \tag{12}$$

For histopathology, reconstructing entire WSIs is infea1sible due to their scale and random patch sampling. Instead, we employ a fixed MLP to map the original patch features $\mathbf{X}_h$ to query embeddings $\mathbf{Q}_h$. The WSI slots $\mathbf{S}_h$ are then decoded back to patch-level embeddings, with cosine similarity enforcing feature-level consistency:

$$\hat{\mathbf{X}}_h = \mathcal{R}(\mathbf{Q}_h, \mathbf{S}_h), \qquad \mathcal{L}_{\text{recon}}^h = 1 - \frac{1}{M_h} \sum_{j=1}^{M_h} \cos(\hat{\mathbf{x}}_{h,j}, \mathbf{x}_{h,j}). \tag{13}$$

**Cross-modal Reconstruction**. To capture molecular-morphological mappings, we introduce a cross-modal reconstruction task in which omics-derived slots are guided to predict pathway embeddings from histopathology. Specifically, WSI patch features are re-encoded through omics-derived slots $\mathbf{S}_g$ and decoded to approximate $\mathbf{X}_g$ with an MSE loss:

$$\tilde{\mathbf{X}}_g = \mathcal{R}(\mathbf{Q}_g, \tilde{\mathbf{S}}_{g \to h}), \qquad \mathcal{L}_{\text{recon}}^{g \to h} = \|\tilde{\mathbf{X}}_g - \mathbf{X}_g\|_2^2. \tag{14}$$

### B.4 DETAILS OF MULTIMODAL INTERACTIONS

In contrast to one-shot co-attention methods (Chen et al., 2021b; Xu & Chen, 2023; Jaume et al., 2024), we employ an *iterative cross-attention* scheme that reuses a single fusion layer for $L$ iterations of bidirectional updates. At each iteration $l$, omics- and histology-derived slots alternately attend to one another, absorbing complementary information while preserving their own context via recurrent updates, akin to slot attention. This iterative design supports multi-step fusion, progressively aligning molecular and morphological evidence that may require hierarchical inference (e.g., molecular dysregulation propagating through cellular morphology to tissue-level structure (Huang et al., 2025)). Concretely, at iteration $l$, the update of histology slots attending to omics slots is defined as

$$\mathbf{A}_{h \leftarrow g}^{(l)} = \text{softmax}\left( \frac{\mathcal{Q}(\hat{\mathbf{S}}_h^{(l)}) \mathcal{K}(\hat{\mathbf{S}}_g^{(l)})^\top}{\sqrt{d}} \right) \in \mathbb{R}^{S_h \times S_g}, \quad \mathbf{U}_h^{(l)} = \mathbf{A}_{h \leftarrow g}^{(l)} \mathcal{V}(\hat{\mathbf{S}}_g^{(l)}),$$
$$\hat{\mathbf{S}}_h^{(l+1)} = \text{RNN}(\hat{\mathbf{S}}_h^{(l)}, \mathbf{U}_h^{(l)}), \quad \hat{\mathbf{S}}_h^{(l+1)} \leftarrow \hat{\mathbf{S}}_h^{(l+1)} + \text{MLP}(\hat{\mathbf{S}}_h^{(l+1)}), \tag{15}$$

with a symmetric update for omics slots attending to histology slots. Repeating for $l = \{0, \dots, L-1\}$ yields progressively aligned multimodal slot representations.

In parallel with cross-attention, each modality undergoes intra-modal self-attention, yielding refined representations $\bar{\mathbf{S}}_h$ and $\bar{\mathbf{S}}_g$. To form the final multimodal representation, we compute pooled summaries (mean over slots) and concatenate three branches outputs:

$$\mathbf{z} = \left[ \text{Pool}([\hat{\mathbf{S}}_h \| \hat{\mathbf{S}}_g]) \, \| \, \text{Pool}(\bar{\mathbf{S}}_h) \, \| \, \text{Pool}(\bar{\mathbf{S}}_g) \right] \in \mathbb{R}^{3d}, \tag{16}$$

where $\|$ denotes concatenation along the slot dimension and $\text{Pool}(\cdot)$ indicates mean pooling over slots. Finally, a prediction head maps $\mathbf{z}$ to discrete risk classes, with training jointly optimizing the survival loss and the auxiliary reconstruction losses described above.

### B.5 DETAILS OF OVERALL LOSS

The total training objective of SlotSPE is

$$\mathcal{L} = \mathcal{L}_{\text{surv}} + \lambda \mathcal{L}_{\text{recon}}. \tag{17}$$

where $\lambda$ balances the survival and reconstruction components. The survival loss $\mathcal{L}_{\text{surv}}$ consists of the survival prediction loss together with each single-modal survival losses. The single-modal survival

predictions, i.e., $y_h$ (histology) and $y_g$ (genomics) in Eq. 18, are obtained by applying their modal-specific MoE decoders to the corresponding slot representations, $\mathbf{S}_h$ and $\mathbf{S}_g$. These single-modal survival losses incorporate task supervision for the gated slot logits $y_h$ and $y_g$:

$$\mathcal{L}_{\mathrm{surv}} = \mathcal{L}_{\mathrm{surv}}(\mathbf{z}, t, c) + \mathcal{L}_{\mathrm{surv}}^h(y_h, t, c) + \mathcal{L}_{\mathrm{surv}}^g(y_g, t, c) \tag{18}$$

while $\mathcal{L}_{\mathrm{recon}}$ combines the slots regularization and cross-modal reconstruction objectives:

$$\mathcal{L}_{\mathrm{recon}} = \mathcal{L}_{\mathrm{recon}}^g + \mathcal{L}_{\mathrm{recon}}^h + \mathcal{L}_{\mathrm{recon}}^{g \to h} \tag{19}$$

## C  DATASETS AND IMPLEMENTATIONS

**Dataset**. We perform comprehensive evaluations across ten publicly available cancer cohorts from TCGA[3] , including Breast Invasive Carcinoma (BRCA), Bladder Urothelial Carcinoma (BLCA), Colon and Rectum Adenocarcinoma (COADREAD), Stomach Adenocarcinoma (STAD), Head and Neck Squamous Cell Carcinoma (HNSC), as well as Lung Adenocarcinoma (LUAD), Lung Squamous Cell Carcinoma (LUSC), Kidney Renal Clear Cell Carcinoma (KIRC), and Skin Cutaneous Melanoma (SKCM). Case counts for each dataset are listed in Table 1. Following  (Jaume et al., 2024), we predict disease-specific survival (DSS), a more precise indicator of patient status than overall survival. Histological data include all diagnostic WSIs, while transcriptomic profiles with DSS labels are obtained from cBioPortal[4]. Pathway gene sets are curated from Hallmarks  (Subramanian et al., 2005; Liberzon et al., 2015) and Reactome  (Gillespie et al., 2022), with genes absent in cBioPortal removed, yielding 330 pathways.

**Evaluation**. We adopt 5-fold cross-validation and report the concordance index (C-index) (Harrell Jr et al., 1996) together with the standard deviation (std) across folds. To further assess clinical relevance, we visualize Kaplan–Meier (KM) survival curves (Kaplan & Meier, 1958) for stratified risk groups and apply the log-rank test (Mantel et al., 1966) to evaluate global differences in survival distributions. In addition, we compute the restricted mean survival time (RMST) (Irwin, 1949; Karrison, 1986). RMST, defined as the area under the estimated survival curve up to a clinically meaningful truncation time (60 months in our experiments), provides an interpretable summary of the average survival time within the specified horizon and does not rely on the proportional hazards assumption. We report: (a) the RMST values for both high- and low-risk groups, (b) the RMST difference ($\Delta$ = High–Low) with a 95% bootstrap confidence interval (CI) and a two-sided $p$-value testing whether the difference is significantly different from zero, and (c) the RMST ratio (High/Low) with a log-transformed 95% bootstrap CI as a scale-free measure of group separation. To ensure robust inference, both CI and $p$-values are estimated from 1000 bootstrap replicates (Tibshirani & Efron, 1993). These statistics are annotated on the KM plots, providing an interpretable and robust evaluation of prognostic stratification.

Taken together, the three metrics are complementary: the C-index measures overall ranking concordance of predicted risks, the log-rank test evaluates survival curve separation under proportional hazards, and RMST provides a clinically grounded and assumption-light measure of absolute and relative survival differences.

**Implementation**. Tissue regions are segmented from WSIs, and non-overlapping $256 \times 256$ patches are extracted at $20\times$ magnification. We adopt UNI  (Chen et al., 2024) as the patch encoder and SNNs as the pathway encoder. The method is implemented in Python using PyTorch and executed on only one NVIDIA A100 GPU. For survival prediction, disease-specific survival (DSS) time is discretized into four intervals. Bag features are embedded via an MLP with a 512-dimensional hidden layer and a 256-dimensional output, thus $d$ and $d_{slot}$ are all set to 256. The number of iterations for slot attention is set to $T = 10$, and for iterative cross-attention to $L = 3$. The top-$K$ in selective slot activation decoder is set to $25\%$ slot numbers since patients are divided into 4 risk groups. We report the best performance by ablating the slot numbers in Appendix F.3.1. Pathway tokens are reconstructed using MSE loss, while histopathology tokens are optimized with cosine similarity loss using a hyperparameter $\lambda = 0.1$ (searched from $[0.01, 0.1, 1.0]$). To support batch training, 4096 patches are randomly sampled per WSI, whereas all patches are used during inference. Optimization is carried out with Adam  (Kingma & Ba, 2014) at a learning rate of $5 \times 10^{-4}$. All models are trained for 30 epochs with a batch size of 32.

---

[3]https://portal.gdc.cancer.gov/
[4]https://www.cbioportal.org/

Table 4: Comparison of C-index (mean ± std) across ten cancer datasets using **biological pathways**. Best and second-best results are in **bold** and underline.

| Model | Modality | BRCA (N=1046) | COADREAD (N=573) | KIRC (N=488) | UCEC (N=488) | LUAD (N=467) | Overall |
|---|---|---|---|---|---|---|---|
| MLP | g. | 0.644± 0.097 | 0.661± 0.053 | 0.750± 0.035 | 0.731± 0.071 | 0.634± 0.069 | - |
| SNN | g. | 0.645±0.086 | 0.664±0.021 | 0.754± 0.035 | 0.753± 0.071 | 0.652± 0.063 | - |
| SNNTrans | g. | 0.657±0.060 | 0.710±0.053 | 0.747±0.035 | 0.724±0.061 | 0.648±0.059 | - |
| **SlotSPE** | g. | 0.698±0.046 | 0.733±0.045 | 0.774±0.022 | 0.752±0.065 | 0.646±0.064 | - |
| ABMIL | h. | 0.656±0.037 | 0.715±0.029 | 0.782±0.026 | 0.779±0.057 | 0.645±0.061 | - |
| TransMIL | h. | 0.647±0.032 | 0.723±0.027 | 0.767±0.035 | 0.759±0.059 | 0.655±0.052 | - |
| CLAM-SB | h. | 0.696±0.049 | 0.742±0.046 | 0.784±0.033 | 0.757±0.072 | 0.639±0.059 | - |
| CLAM-MB | h. | 0.715±0.034 | 0.721±0.041 | 0.782±0.014 | 0.788±0.074 | 0.632±0.070 | - |
| **SlotSPE** | h. | 0.738±0.019 | 0.729±0.038 | 0.790±0.023 | 0.779±0.057 | 0.656±0.038 | - |
| Porpoise | g.+h. | 0.651±0.046 | 0.699±0.050 | 0.788±0.028 | 0.788±0.092 | 0.647±0.091 | - |
| MCAT | g.+h. | 0.709±0.052 | 0.737±0.033 | 0.795±0.018 | 0.802±0.060 | 0.659±0.054 | - |
| MOTCat | g.+h. | 0.717±0.064 | 0.742±0.042 | 0.799±0.013 | 0.773±0.060 | 0.663±0.052 | - |
| CMTA | g.+h. | 0.731±0.066 | 0.720±0.008 | 0.794±0.018 | 0.764±0.069 | 0.662±0.053 | - |
| SurvPath | g.+h. | 0.728±0.040 | 0.727±0.024 | 0.787±0.025 | 0.751±0.084 | 0.661±0.050 | - |
| PIBD | g.+h. | 0.662±0.054 | 0.716±0.054 | 0.772±0.007 | 0.783±0.053 | 0.634±0.065 | - |
| MMP | g.+h. | 0.714 ± 0.050 | 0.686 ± 0.016 | 0.770 ±0.011 | 0.755 ±0.073 | 0.634 ± 0.048 | - |
| LD-CVAE | g.+h. | 0.705±0.050 | 0.753±0.029 | 0.792±0.025 | 0.788±0.068 | 0.651±0.055 | - |
| **SlotSPE** (8 slots/modality) | g.+h. | 0.748±0.052 | 0.759±0.027 | 0.802±0.009 | 0.807±0.067 | 0.653±0.056 | - |
| **SlotSPE** ($T=3$) | g.+h. | 0.732±0.056 | 0.763±0.031 | **0.817±0.020** | 0.788±0.067 | 0.672±0.056 | - |
| **SlotSPE** | g.+h. | **0.779±0.061** | **0.773±0.042** | 0.815±0.017 | **0.813±0.059** | **0.683±0.058** | - |

| Model | Modality | LUSC (N=460) | HNSC (N=438) | SKCM (N=409) | BLCA (N=381) | STAD (N=366) | Overall |
|---|---|---|---|---|---|---|---|
| MLP | g. | 0.584±0.072 | 0.584±0.061 | 0.652±0.038 | 0.677±0.053 | 0.615±0.063 | 0.653 |
| SNN | g. | 0.548±0.041 | 0.597±0.039 | 0.673±0.021 | 0.694±0.040 | 0.585±0.048 | 0.656 |
| SNNTrans | g. | 0.555±0.042 | 0.605±0.065 | 0.684±0.029 | 0.699±0.031 | 0.593±0.038 | 0.662 |
| **SlotSPE** | g. | 0.616±0.058 | 0.611±0.038 | 0.678±0.025 | 0.701±0.017 | 0.596±0.036 | 0.681 |
| ABMIL | h. | 0.593±0.055 | **0.664±0.043** | 0.668±0.042 | 0.663±0.072 | 0.642±0.044 | 0.681 |
| TransMIL | h. | 0.582±0.039 | 0.644±0.020 | 0.632±0.050 | 0.647±0.032 | 0.656±0.038 | 0.671 |
| CLAM-SB | h. | 0.584±0.065 | 0.649±0.042 | 0.643±0.049 | 0.645±0.052 | 0.637±0.022 | 0.678 |
| CLAM-MB | h. | 0.608±0.056 | 0.636±0.038 | 0.640±0.032 | 0.657±0.029 | 0.641±0.032 | 0.682 |
| **SlotSPE** | h. | 0.598±0.054 | 0.635±0.038 | 0.659±0.063 | 0.642±0.035 | **0.676±0.033** | 0.690 |
| Porpoise | g.+h. | 0.597±0.050 | 0.599±0.048 | 0.682±0.021 | 0.690±0.025 | 0.619±0.022 | 0.676 |
| MCAT | g.+h. | 0.583±0.051 | 0.622±0.049 | 0.667±0.049 | 0.688±0.024 | 0.639±0.032 | 0.690 |
| MOTCat | g.+h. | 0.573±0.026 | 0.627±0.049 | **0.688±0.054** | 0.700±0.010 | 0.642±0.019 | 0.692 |
| CMTA | g.+h. | 0.599±0.071 | 0.619±0.051 | 0.681±0.036 | 0.702±0.033 | 0.639±0.038 | 0.691 |
| SurvPath | g.+h. | 0.577±0.024 | 0.611±0.024 | 0.666±0.054 | 0.666±0.034 | 0.647±0.027 | 0.682 |
| PIBD | g.+h. | 0.575±0.098 | 0.627±0.069 | 0.649±0.048 | 0.693±0.032 | 0.644±0.038 | 0.676 |
| MMP | g.+h. | 0.580 ± 0.043 | 0.580 ± 0.049 | 0.682 ± 0.029 | 0.686 ± 0.004 | 0.602 ± 0.020 | 0.669 |
| LD-CVAE | g.+h. | 0.620±0.045 | 0.635±0.051 | 0.682±0.037 | 0.635±0.037 | 0.653±0.036 | 0.692 |
| **SlotSPE** (8 slots/modality) | g.+h. | 0.620±0.058 | 0.642±0.035 | **0.688±0.051** | 0.705±0.037 | 0.637±0.018 | 0.706 |
| **SlotSPE** ($T=3$) | g.+h. | 0.610±0.064 | 0.624±0.052 | 0.676±0.056 | 0.703±0.023 | 0.657±0.048 | 0.704 |
| **SlotSPE** | g.+h. | **0.634±0.079** | 0.642±0.035 | **0.688±0.051** | **0.708±0.025** | 0.671±0.028 | **0.721** |

## D  COMPARISONS WITH SOTAS

The complete results with standard deviations (std) for Table 1, using the biological pathway format for gene data, are provided in Table 4. In addition, we also report results using 8 slots per modality and $T = 3$ iterations in slot attention, with further ablations on the number of slots provided in Appendix F.3.1 and on the number of iterations in Appendix F.3.2. The results demonstrate that even with a very small number of slots for both histopathology and genomics, SlotSPE maintains strong performance, indicating that the slots effectively capture structured prognostic events through highly compressed representations.

To further assess the generalization ability of our approach, we compare SlotSPE with other multi-modal methods under a higher-level grouping of genes into six functional categories (Subramanian

Table 5: Comparison of C-index (mean $\pm$ std) across ten cancer datasets using **six gene functional groups**. Best and second-best results are in **bold** and underline.

| Model | Modality | BRCA (N=1046) | COADREAD (N=573) | KIRC (N=488) | UCEC (N=488) | LUAD (N=467) | Overall |
|---|---|---|---|---|---|---|---|
| Porpoise | g.+h. | 0.681±0.075 | 0.742±0.038 | 0.778±0.023 | 0.798±0.086 | 0.655±0.066 | - |
| MCAT | g.+h. | 0.706±0.038 | 0.736±0.024 | 0.797±0.018 | 0.800±0.065 | 0.641±0.062 | - |
| MOTCat | g.+h. | 0.683±0.031 | 0.759±0.039 | 0.791±0.010 | 0.808±0.041 | **0.683±0.057** | - |
| CMTA | g.+h. | 0.670±0.031 | 0.751±0.031 | 0.786±0.029 | 0.779±0.059 | 0.644±0.031 | - |
| SurvPath | g.+h. | 0.707±0.049 | 0.751±0.029 | 0.768±0.010 | 0.764±0.072 | 0.663±0.050 | - |
| PIBD | g.+h. | 0.683±0.016 | 0.733±0.030 | 0.791±0.019 | 0.774±0.054 | 0.649±0.077 | - |
| MMP | g.+h. | 0.684 ±0.081 | 0.699 ± 0.032 | 0.767±0.017 | 0.772±0.056 | 0.643±0.041 | - |
| LD-CVAE | g.+h. | 0.695±0.043 | 0.736±0.029 | 0.797±0.025 | 0.792±0.052 | 0.677±0.050 | - |
| **SlotSPE** | g.+h. | **0.735±0.051** | **0.780±0.044** | **0.804±0.020** | **0.824±0.062** | 0.669±0.063 | - |

| Model | Modality | LUSC (N=460) | HNSC (N=438) | SKCM (N=409) | BLCA (N=381) | STAD (N=366) | Overall |
|---|---|---|---|---|---|---|---|
| Porpoise | g.+h. | 0.585±0.065 | 0.617±0.058 | 0.674±0.035 | 0.683±0.040 | 0.620±0.036 | 0.683 |
| MCAT | g.+h. | 0.571±0.056 | **0.658±0.025** | 0.679±0.041 | 0.682±0.029 | 0.630±0.011 | 0.690 |
| MOTCat | g.+h. | 0.578±0.053 | 0.634±0.036 | **0.688±0.049** | 0.689±0.045 | 0.657±0.007 | 0.692 |
| CMTA | g.+h. | 0.582±0.048 | 0.651±0.056 | 0.667±0.031 | 0.682±0.039 | 0.627±0.045 | 0.684 |
| SurvPath | g.+h. | 0.564±0.033 | 0.636±0.036 | 0.658±0.064 | **0.707±0.049** | 0.637±0.021 | 0.686 |
| PIBD | g.+h. | 0.595±0.100 | 0.626±0.029 | 0.649±0.037 | 0.676±0.058 | 0.648±0.044 | 0.682 |
| MMP | g.+h. | 0.584±0.048 | 0.588±0.054 | 0.667±0.034 | 0.680±0.051 | 0.597±0.029 | 0.668 |
| LD-CVAE | g.+h. | 0.613±0.051 | 0.633±0.035 | 0.645±0.040 | 0.684±0.043 | 0.663±0.041 | 0.693 |
| **SlotSPE** | g.+h. | **0.616±0.049** | 0.645±0.030 | 0.679±0.035 | 0.701±0.026 | **0.667±0.017** | **0.712** |

et al., 2005; Liberzon et al., 2015; Chen et al., 2021b; Xu & Chen, 2023; Zhou & Chen, 2023): tumor supression, oncogenesis, protein kinases, cellular differentiation, transcription, and cytokines and growth. In this setting, the number of gene slots is fixed to six, while the number of histopathology slots is varied and the best performance is reported. Results in Table 5 show that SlotSPE achieves the best overall performance, ranking first or second in 8 out of 10 cohorts, and outperforming the second-best method (LD-CVAE) by 1.9% on average. This indicates that even when genes are grouped into higher-level functional categories, our SlotSPE is able to maintain strong predictive performance across different granularities of gene input.

## E  ROBUSTNESS AND RISK STRATIFICATION

The complete results with standard deviations for Table 2, using the biological pathway representation for gene data, are reported in Table 6. While most baselines are not explicitly designed to handle missing modalities, they can still be evaluated under missing-genomics settings by supplying a neutral placeholder input in place of genomic features (Zhang et al., 2025). For completeness, we also include two strong methods (MOTCAT and CMTA). The results show that when these approaches rely solely on histology—without modeling or reconstructing genomic information—their prognostic performance drops substantially. Moreover, under missing-genomics conditions, their limited robustness leads to performance even worse than strong single-modal baselines (e.g., ABMIL).

Kaplan–Meier curves for all ten datasets are shown in Figure 6. For clarity, Table 7 summarizes four complementary statistics: (a) log-rank $p$-values across all cancer types, (b) RMST values for both high- and low-risk groups, (c) the RMST difference ($\Delta$ = High–Low) with 95% confidence intervals, and (d) the RMST ratio (High/Low) with 95% confidence intervals. Across most datasets, SlotSPE consistently achieves larger RMST differences (Because $\Delta$ is typically negative, a "larger" separation corresponds to a more negative value, i.e. smaller numerically but larger in magnitude) and smaller RMST ratios than competing methods, demonstrating clearer prognostic separation. Even in datasets where performance is comparable (e.g., KIRC, LUSC, HNSC), SlotSPE generally maintains equal or better significance with tighter confidence intervals.

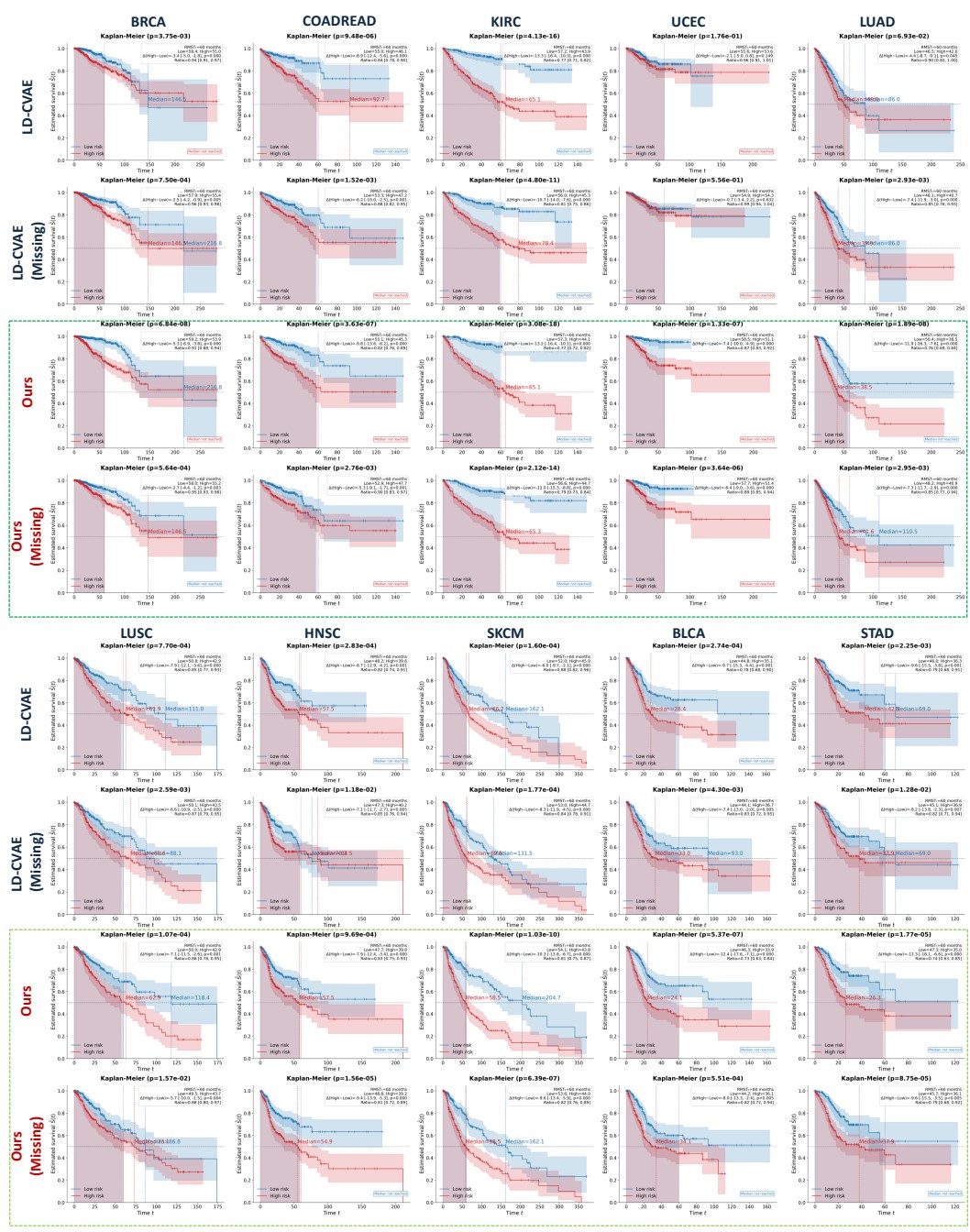

Figure 6: Kaplan–Meier curves of predicted high-risk and low-risk groups. A $p$-value $< 0.05$ at the top indicates statistically significant separation between groups. The restricted mean survival time (RMST) up to 60 months is also reported, with values shown as $\Delta$ (High–Low), and RMST ratio (High/Low). (Zoom in to view details.)

Table 6: C-index (mean $\pm$ std) across ten cancer datasets under missing-genomics settings. The column *Missing* indicates whether the genomic modality is missing ($\checkmark$ = missing). Best and second-best results are highlighted in **bold** and underline.

| Model | Missing | BRCA (N=1046) | COADREAD (N=573) | KIRC (N=488) | UCEC (N=488) | LUAD (N=467) | Overall |
|---|---|---|---|---|---|---|---|
| MOTCAT | $\checkmark$ | 0.600±0.077 | 0.645±0.028 | 0.738±0.034 | 0.659±0.144 | 0.643±0.023 | - |
| CMTA | $\checkmark$ | 0.604±0.112 | 0.683±0.065 | 0.662±0.045 | 0.676±0.124 | 0.593±0.051 | - |
| LD-CVAE | | 0.705±0.050 | 0.753±0.029 | 0.792±0.025 | 0.788±0.068 | 0.651±0.055 | - |
| LD-CVAE | $\checkmark$ | 0.715±0.056 | 0.736±0.042 | 0.798±0.014 | 0.775±0.073 | 0.638±0.016 | - |
| SlotSPE | | **0.779±0.061** | **0.773±0.042** | **0.815±0.018** | **0.813±0.059** | **0.683±0.057** | - |
| SlotSPE | $\checkmark$ | 0.734±0.056 | 0.741±0.040 | 0.789±0.029 | 0.802±0.047 | 0.663±0.054 | - |

| Model | Missing | LUSC (N=460) | HNSC (N=438) | SKCM (N=409) | BLCA (N=381) | STAD (N=366) | Overall |
|---|---|---|---|---|---|---|---|
| MOTCAT | $\checkmark$ | 0.540±0.024 | 0.578±0.030 | 0.642±0.050 | 0.565±0.098 | 0.585±0.071 | 0.619 |
| CMTA | $\checkmark$ | 0.541±0.045 | 0.564±0.074 | 0.541±0.048 | 0.558±0.054 | 0.608±0.065 | 0.603 |
| LD-CVAE | | 0.620±0.045 | 0.635±0.051 | 0.682±0.037 | 0.635±0.037 | 0.653±0.036 | 0.692 |
| LD-CVAE | $\checkmark$ | 0.596±0.042 | **0.649±0.049** | 0.676±0.038 | 0.644±0.048 | 0.647±0.049 | 0.688 |
| SlotSPE | | **0.634±0.079** | 0.642±0.035 | **0.688±0.051** | **0.708±0.025** | 0.671±0.028 | **0.721** |
| SlotSPE | $\checkmark$ | **0.634±0.032** | **0.649±0.031** | 0.678±0.068 | 0.678±0.041 | **0.677±0.029** | 0.704 |

Table 7: Summary of log-rank $p$-values and RMST-based statistics (RMST values, differences, and ratios) for group separation across ten cancer datasets in Figure 6.

| Dataset | Method | Log-rank $p$-value | RMST (Low) | RMST (High) | $\Delta$ (High–Low) [95% CI] | Ratio (High/Low) [95% CI] |
|---|---|---|---|---|---|---|
| BRCA | LD-CVAE | $3.75 \times 10^{-3}$ | 58.4 | 55.0 | -3.4 [-5.0, -1.8] | 0.94 [0.91, 0.97] |
| | SlotSPE | $6.84 \times 10^{-8}$ | 59.2 | 53.9 | -5.3 [-6.9, -3.8] | 0.91 [0.88, 0.94] |
| COADREAD | LD-CVAE | $9.48 \times 10^{-6}$ | 55.0 | 46.1 | -8.9 [-12.4, -5.6] | 0.84 [0.78, 0.90] |
| | SlotSPE | $3.63 \times 10^{-7}$ | 55.1 | 45.3 | -9.8 [-13.6, -6.2] | 0.82 [0.76, 0.89] |
| KIRC | LD-CVAE | $4.13 \times 10^{-16}$ | 57.2 | 43.9 | -13.3 [-16.4, -10.3] | 0.77 [0.71, 0.82] |
| | SlotSPE | $3.08 \times 10^{-18}$ | 57.3 | 44.1 | -13.2 [-16.4 -10.1] | 0.77 [0.72, 0.82] |
| UCEC | LD-CVAE | $1.76 \times 10^{-1}$ | 55.6 | 53.6 | -2.1 [-5.0, 0.8] | 0.96 [0.91, 1.00] |
| | SlotSPE | $1.33 \times 10^{-7}$ | 58.5 | 51.1 | -7.4 [-10.0, -4.9] | 0.87 [0.83, 0.92] |
| LUAD | LD-CVAE | $6.93 \times 10^{-2}$ | 46.5 | 42.0 | -4.4 [-8.7, -0.1] | 0.90 [0.82, 1.00] |
| | SlotSPE | $1.89 \times 10^{-8}$ | 50.4 | 38.5 | -11.9 [-16.3 -7.6] | 0.76 [0.69, 0.84] |
| LUSC | LD-CVAE | $7.70 \times 10^{-4}$ | 50.8 | 42.9 | -7.9 [-12.1, -3.6] | 0.85 [0.77, 0.93] |
| | SlotSPE | $1.07 \times 10^{-4}$ | 50.0 | 42.9 | -7.1 [-11.5, -2.6] | 0.86 [0.78, 0.95] |
| HNSC | LD-CVAE | $2.83 \times 10^{-4}$ | 48.2 | 39.6 | -8.7 [-12.9, -4.2] | 0.82 [0.74, 0.91] |
| | SlotSPE | $9.69 \times 10^{-4}$ | 47.7 | 39.8 | -7.9 [-12.4, -3.4] | 0.83 [0.75, 0.93] |
| SKCM | LD-CVAE | $1.60 \times 10^{-4}$ | 52.0 | 45.9 | -6.0 [-9.7, -2.1] | 0.88 [0.82, 0.96] |
| | SlotSPE | $1.03 \times 10^{-10}$ | 54.1 | 43.8 | -10.3 [-13.8, -6.7] | 0.81 [0.75, 0.87] |
| BLCA | LD-CVAE | $2.74 \times 10^{-4}$ | 44.8 | 35.1 | -9.7 [-15.3, -4.4] | 0.78 [0.68, 0.90] |
| | SlotSPE | $5.37 \times 10^{-7}$ | 46.3 | 33.9 | -12.4 [-17.6, -7.1] | 0.73 [0.63, 0.84] |
| STAD | LD-CVAE | $2.25 \times 10^{-3}$ | 46.0 | 36.3 | -9.6 [-15.5, -3.8] | 0.79 [0.68, 0.91] |
| | SlotSPE | $1.77 \times 10^{-5}$ | 47.3 | 35.0 | -12.3 [-18.1, -6.6] | 0.74 [0.63, 0.85] |

# F  ABLATION STUDY

## F.1  COMPONENT VALIDATION

The complete results with standard deviations for Table 3, using the biological pathway representation for gene data, are reported in Table 8.

## F.2  HISTOLOGY ENCODER ABLATIONS

To assess the robustness of SlotSPE to the choice of visual encoder, we first replace the histology encoder with a ResNet50 (Srivastava et al., 2015; He et al., 2016) pretrained only on ImageNet (Deng et al., 2009), rather than a pathology-specific foundation model. This setting tests whether SlotSPE

Table 8: Ablation of model components reported as C-index (mean ± std) across ten cancer datasets. Best and second-best results are in **bold** and underline.

| Variants | BRCA (N=1046) | COADREAD (N=573) | KIRC (N=488) | UCEC (N=488) | LUAD (N=467) | Overall |
|---|---|---|---|---|---|---|
| Baseline | 0.711±0.052 | 0.734±0.021 | 0.784±0.010 | 0.779±0.081 | 0.661±0.059 | - |
| w/o Selective Slot Attention | 0.728±0.057 | 0.755±0.030 | 0.787±0.011 | 0.800±0.064 | 0.659±0.067 | - |
| w/o Slots Regularization | 0.734±0.065 | 0.763±0.029 | 0.794±0.017 | 0.811±0.069 | 0.663±0.053 | - |
| w/o Cross-modal Reconstruction | 0.710±0.067 | 0.758±0.027 | 0.787±0.008 | 0.794±0.056 | 0.663±0.062 | - |
| w/o Iterative Cross-attention | 0.727±0.031 | 0.761±0.033 | 0.803±0.027 | 0.794±0.056 | 0.653±0.062 | - |
| SlotSPE | **0.779±0.061** | **0.773±0.042** | **0.815±0.018** | **0.813±0.059** | **0.683±0.057** | - |

| Variants | LUSC (N=460) | HNSC (N=438) | SKCM (N=409) | BLCA (N=381) | STAD (N=366) | Overall |
|---|---|---|---|---|---|---|
| Baseline | 0.592±0.064 | 0.610±0.056 | 0.668±0.043 | 0.686±0.030 | 0.644±0.021 | 0.687 |
| w/o Selective Slot Attention | 0.620±0.071 | 0.615±0.045 | 0.685±0.052 | 0.688±0.042 | 0.650±0.026 | 0.699 |
| w/o Slots Regularization | 0.611±0.060 | 0.621±0.061 | 0.679±0.053 | 0.693±0.038 | 0.667±0.018 | 0.704 |
| w/o Cross-model Reconstruction | 0.619±0.090 | 0.611±0.058 | 0.671±0.026 | 0.698±0.036 | 0.645±0.017 | 0.696 |
| w/o Iterative Cross-attention | 0.602±0.071 | 0.624±0.048 | 0.676±0.039 | 0.702±0.024 | 0.648±0.017 | 0.699 |
| SlotSPE | **0.634±0.079** | **0.642±0.035** | **0.688±0.051** | **0.708±0.025** | **0.671±0.028** | **0.721** |

Table 9: Comparison of C-index (mean ± std) across five datasets using ResNet50 as histology encoder. Best and second-best results are in **bold** and underline.

| Methods | Modality | BRCA (N=1046) | COADREAD (N=573) | KIRC (N=488) | UCEC (N=488) | LUAD (N=467) | Overall |
|---|---|---|---|---|---|---|---|
| TransMIL | g. | 0.608±0.029 | 0.641±0.019 | 0.661±0.036 | 0.690±0.081 | 0.599±0.054 | 0.640 |
| CLAM_MB | g. | 0.573±0.035 | 0.616±0.081 | 0.637±0.076 | 0.655±0.092 | 0.590±0.066 | 0.620 |
| MCAT | g.+h. | 0.694±0.074 | 0.712±0.034 | 0.754±0.023 | 0.771±0.062 | 0.633±0.058 | 0.713 |
| MOTCAT | g.+h. | 0.706±0.062 | 0.674±0.034 | 0.752±0.026 | **0.790±0.053** | 0.637±0.054 | 0.711 |
| SurvPath | g.+h. | 0.650±0.038 | 0.674±0.051 | 0.743±0.037 | 0.761±0.052 | 0.643±0.046 | 0.694 |
| LD-CVAE | g.+h. | 0.656±0.051 | 0.676±0.025 | 0.714±0.061 | 0.733±0.081 | 0.620±0.037 | 0.680 |
| **SlotSPE** | g.+h. | **0.730±0.072** | **0.730±0.035** | **0.769±0.024** | 0.775±0.052 | **0.645±0.064** | **0.730** |

can still achieve strong performance when the visual backbone lacks domain-specific pretraining. For this experiment, we evaluate representative unimodal and multimodal baselines listed in Table 1. As summarized in Table 9, SlotSPE continues to outperform all baselines under this weaker configuration, suggesting that its performance is not tightly coupled to a specialized encoder. We further extend this analysis by examining SlotSPE with recent pathology foundation models, including CONCH (Lu et al., 2024), CONCH v1.5 (used in TITAN (Ding et al., 2025)), and UNI (Chen et al., 2024). The results in Figure 7 show that performance varies across different encoders: while recent pathology foundation models generally improve results compared to ImageNet-pretrained encoder, the best performance is obtained with UNI. These results indicate that SlotSPE is both robust to weaker, non-specialized encoders and compatible with strong pathology-specific foundation models.

## F.3 HYPERPARAMETERS ABLATIONS IN SLOT ATTENTION

### F.3.1 SLOT NUMBERS ABLATIONS

Figure 8 illustrates how the number of slots for histology ($S_h$) and genomics ($S_g$) affects performance across six TCGA cohorts. In most cases, increasing the number of slots usually improves performance compared to the minimal baseline (red line), suggesting that additional slots enable the model to capture a broader set of prognostic events. However, beyond a moderate slot size (e.g., 16–32 slots), further increases often yield little to no gain, and in some cases even slight declines, indicating that excessive slots may introduce redundancy or prognostic irrelevant patterns.

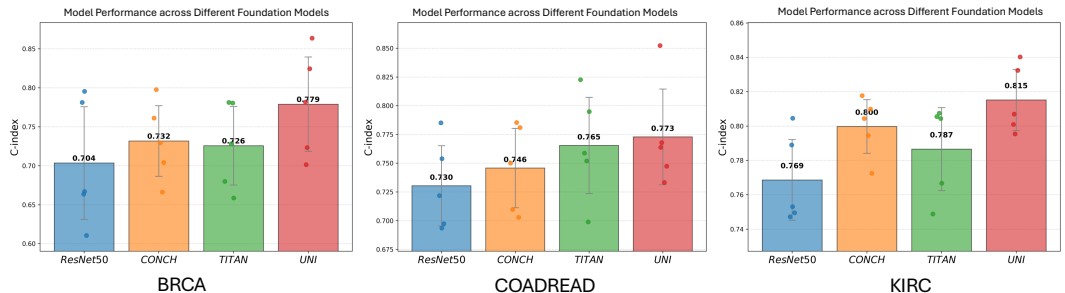

Figure 7: Effect of different foundation models used as histology encoders. Performance (C-index) is shown for three cohorts (BRCA, COADREAD, KIRC) using four backbone encoders: ResNet50, CONCH, TITAN, and UNI. Bars represent mean across five folds, with error bars indicating standard deviation. Higher values indicate better performance.

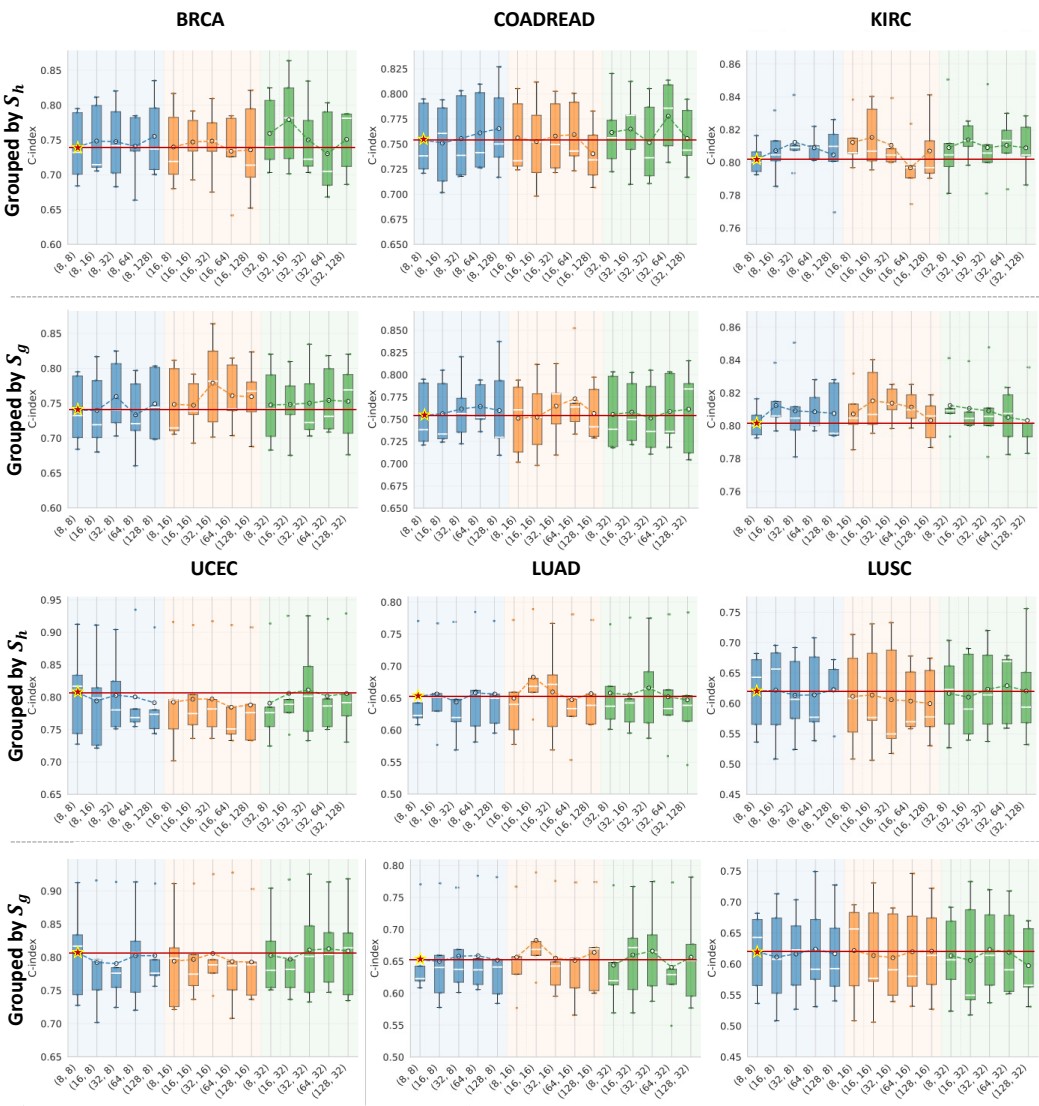

Figure 8: Ablation on slot numbers. Boxplots show C-index across cohorts as the number of histology slots ($S_h$, top) and genomic slots ($S_g$, bottom) varies, with the x-axis indicating the pair ($S_h$, $S_g$). The red line denotes the baseline with the minimum slot setting. Increasing the number of slots generally improves performance, though gains plateau once saturation is reached.

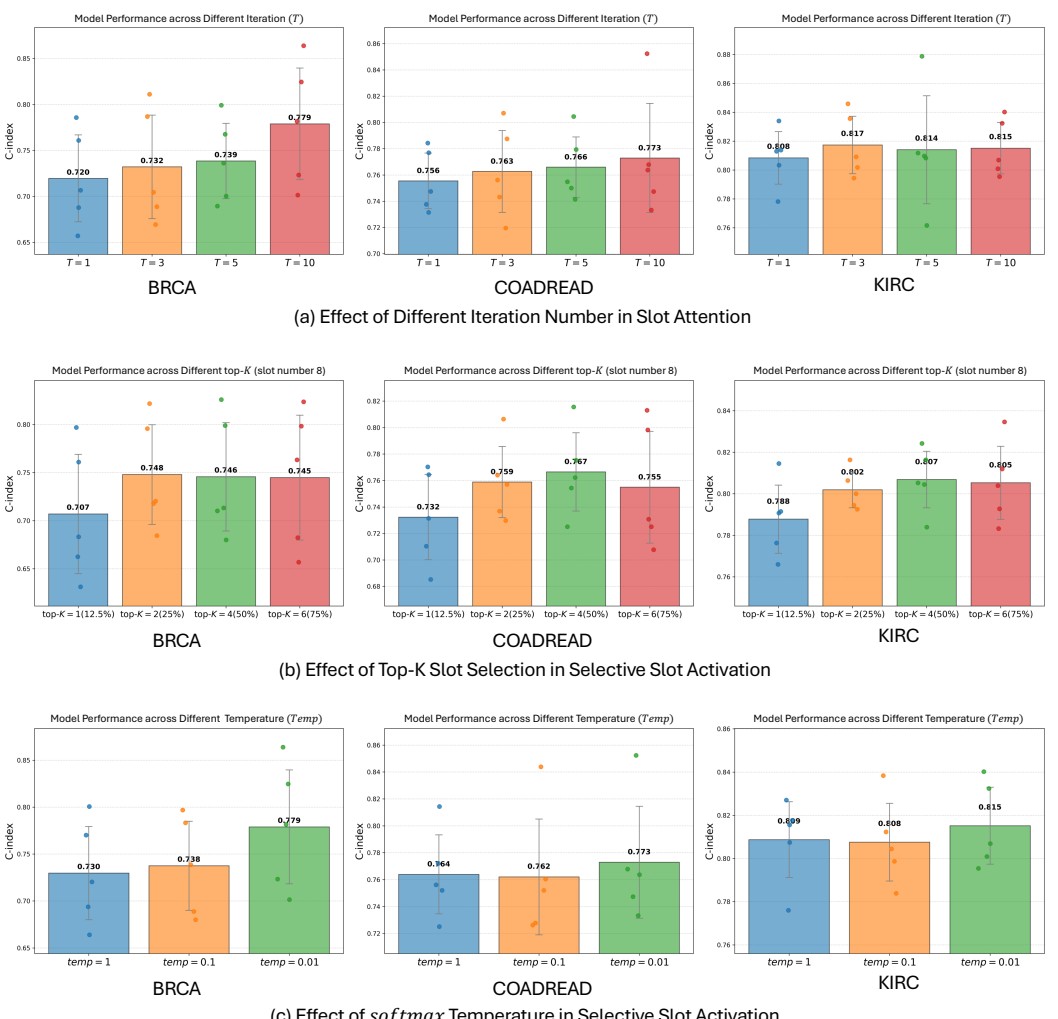

Figure 9: Ablation study of hyperparameters. (a) Effect of different iteration number $T$ in the slot attention. (b) Effect of different top-$K$(percentage of slots) in selective slot activation (slot number is fixed as 8). (c) Effect of $softmax$ temperature on in selective slot activation. Bars represent mean across five folds, with error bars indicating standard deviation. Higher values indicate better performance.

### F.3.2   SLOT ATTENTION ITERATIONS ABLATIONS.

We ablate the number of iterations $T$ in slot attention to examine its impact on performance shown in Figure 9(a). With a small number of iterations ($T = 1$), slots are updated only once, limiting their ability to refine representations and capture complex prognostic patterns, which results in lower C-index values (e.g., 0.720 on BRCA). As $T$ increases, slot updates stabilize, enabling better information bindings and improved outcomes (best at $T = 10$ on BRCA with 0.779). In datasets such as KIRC, performance saturates around $T = 3$–$5$, suggesting diminishing returns after sufficient updates. Although larger $T$ enhances stability and accuracy, it also increases training time. Overall, our results indicate that moderate values provide a favorable trade-off between performance and efficiency. For consistency, we also report main results under the $T = 3$ setting in Table 4.

## F.4 HYPERPARAMETERS ABLATIONS IN SELECTIVE SLOT ACTIVATION

### F.4.1 TOP-$K$ SLOT SELECTION ABLATIONS

We analyze the effect of top-$K$ selection by fixing the slot number to 8 for both modalities and evaluate four settings: top-1 (12.5% all slots), top-2 (25% all slots), top-4 (50% all slots), and top-6 (75% all slots), as shown in Figure 9 (b). Using only top-1 results in the weakest performance, which is expected, because patients are divided into four risk groups, activating only a single slot per patient causes several slots to be rarely activated during training. Conversely, when top-$K$ becomes too large (e.g., top-6), some slots that are not discriminative for a given patient are still forced to activate. While the 25% and 50% settings perform similarly across datasets, we adopt the 25% configuration because it aligns naturally with our design: with four risk groups, selecting roughly one-quarter of the slots provides a trade-off between sparsity and coverage.

### F.4.2 SOFTMAX TEMPERATURE ABLATIONS

We further evaluate the effect of $softmax$ temperature (denoted as $temp$) in selective slot activation as shown in Figure 9 (c). A higher temperature (e.g., $temp = 1$) produces soft, diffuse gating and results in weaker performance. Lowering the temperature sharpens the gating distribution and leads to more discriminative slot selection, generally improving results. The setting $temp = 0.01$ achieves the best C-index across three datasets, and we therefore adopt it in our experiments.

## G EFFICIENCY ANALYSIS

In this section, we analyze the efficiency of our model by comparing it with the four strongest baselines in Table 1 without missing modality setting on identical hardware (An NVIDIA RTX A6000, 48GB). Since all methods use the same WSI and omics encoders, we omit these components and focus our comparison solely on the remaining model modules for a more meaningful efficiency assessment.

### G.1 INFERENCE RUNTIME AND MEMORY

We compare inference (where all patches are used) peak memory and wall-clock time against model performance on the COADREAD dataset. In this comparison, we set our model with slot number as 16 and iteration as 3. As shown in Figure 10(a), SlotSPE achieves the highest C-index while maintaining low runtime and memory usage. MCAT and MOTCAT achieve shorter runtime primarily because they omit self-attention computations within WSI patches; however, this design choice leads to a considerable drop in predictive performance. CMTA improves upon these baselines by employing Nyström attention (Xiong et al., 2021) as an approximation to full self-attention, but its capacity remains insufficient to capture the sparse, patient-specific prognostic signals. LD-CVAE attains the second-best performance, yet its reliance on a variational autoencoder introduces substantial memory and runtime overhead.

### G.2 TRAINING RUNTIME AND MEMORY

We also compare training-time runtime and memory usage. Since the official implementations of MCAT, MOTCAT, and LD-CVAE don't support batch training, we set the batch size to 1 for all methods and randomly sample 4,096 patches to ensure a fair comparison. For SlotSPE, we retain the same configuration of 16 slots and 3 iterations. As shown in Figure 10(b), SlotSPE exhibits moderately higher memory usage and training time compared with its inference cost, primarily due to the additional reconstruction component that is active during training. Nonetheless, its overall training resource footprint remains well balanced and sits near the middle of all evaluated methods.

### G.3 MODULE-LEVEL COST BREAKDOWN

To identify the main computational bottlenecks, Figure 10(c) reports the module-level breakdown of GPU memory consumption and runtime within SlotSPE. The reconstruction component dominates both memory (45.0%) and runtime (55.4%), accounting for the largest share of training cost. Slot attention ranks second, contributing 26.2% of memory usage and 36.9% of runtime. The remaining modules—including the MoE-style decoder, slot interaction, and the final predictor—together account for a relatively small proportion of the overall cost. The heavier cost of the reconstruction branch arises from the inclusion of two slot regularizations for both modalities and one cross-modal reconstruction. These reconstruction components are used only during training and are disabled at inference unless the omics modality is missing, in which case the cross-modal reconstruction head is activated.

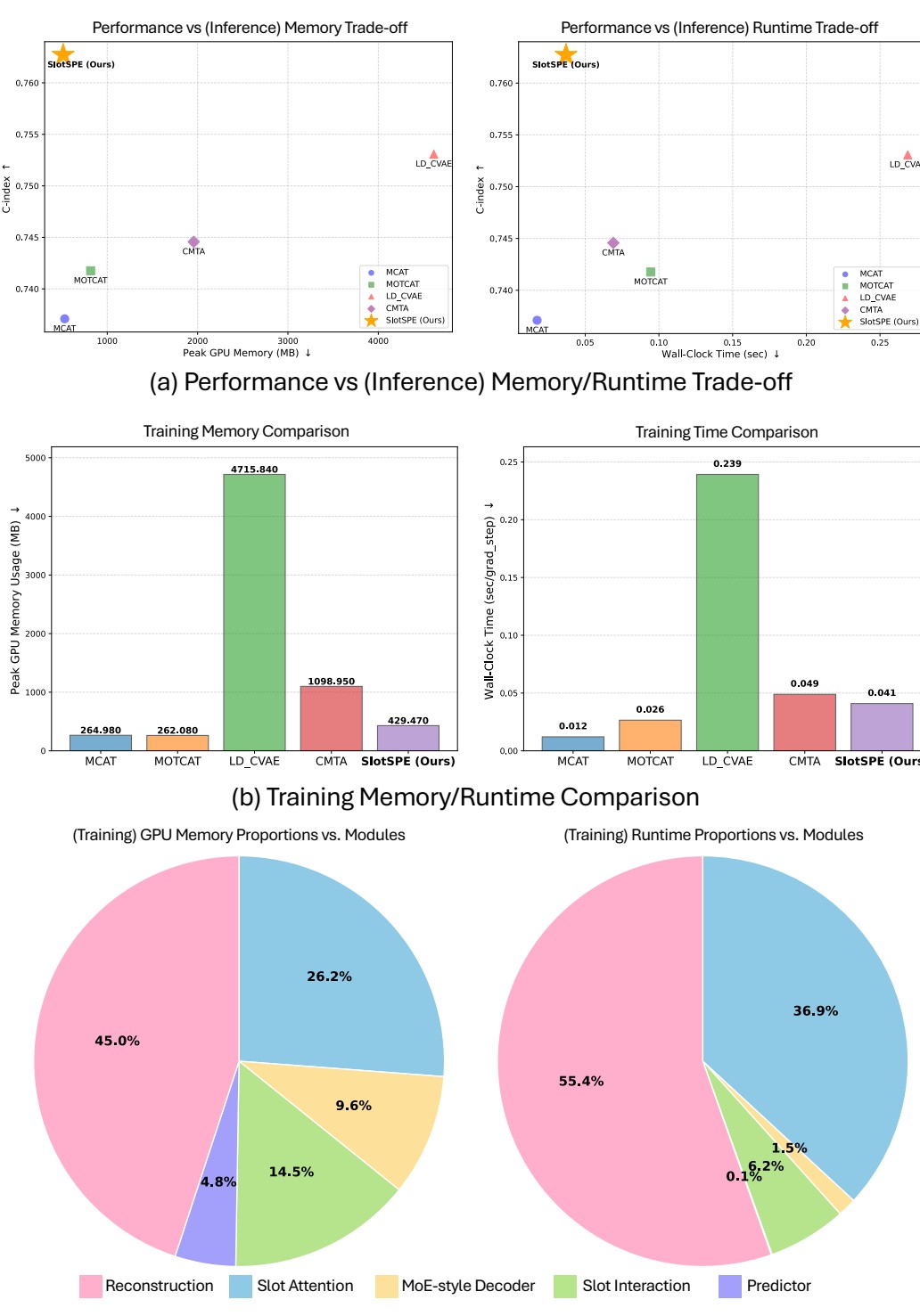

Figure 10: Efficiency analysis. (a) Performance vs. inference memory/runtime trade-off across compared models. (b) Training memory/runtime comparison under a unified batch-size setting. (c) Module-level breakdown of training GPU memory/runtime proportions.

Table 10: Multivariable Cox analysis with clinical variables and SlotSPE predictions. Comparison of clinical-only vs. clinical + model configurations using AUC, IBS, C-index, and ΔC-index across five cohorts. The clinical variables of the TCGA cohort were downloaded from cBioPortal.

| Cohort | AUC ↑ | | IBS ↓ | | C-index ↑ | | ΔC-index |
|---|---|---|---|---|---|---|---|
| | clinical | clinical+model | clinical | clinical+model | clinical | clinical+model | |
| BRCA | 0.731±0.051 | 0.782±0.066 | 0.134±0.021 | 0.133±0.022 | 0.768±0.058 | 0.816±0.071 | 0.053±0.064 |
| COADREAD | 0.778±0.053 | 0.782±0.062 | 0.138±0.017 | 0.140±0.030 | 0.780±0.072 | 0.819±0.058 | 0.040±0.022 |
| KIRC | 0.858±0.049 | 0.889±0.039 | 0.137±0.020 | 0.110±0.021 | 0.844±0.013 | 0.868±0.015 | 0.023±0.014 |
| LUAD | 0.722±0.114 | 0.726±0.085 | 0.176±0.008 | 0.174±0.011 | 0.683±0.080 | 0.739±0.061 | 0.054±0.044 |
| LUSC | 0.681±0.084 | 0.747±0.111 | 0.185±0.018 | 0.166±0.030 | 0.652±0.059 | 0.699±0.077 | 0.047±0.057 |

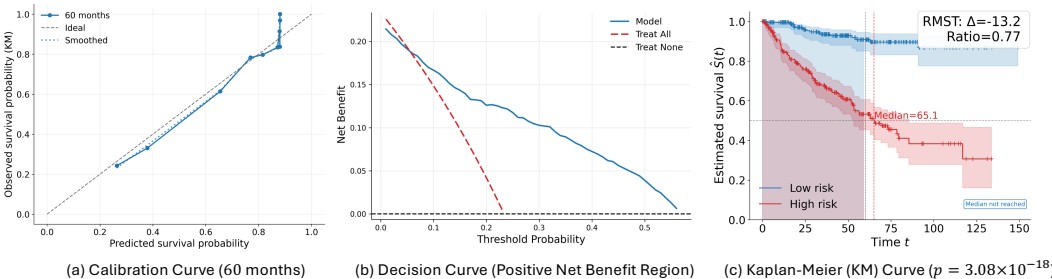

(a) Calibration Curve (60 months)    (b) Decision Curve (Positive Net Benefit Region)    (c) Kaplan–Meier (KM) Curve ($p = 3.08 \times 10^{-18}$)

Figure 11: Calibration curve, decision curve analysis, and Kaplan–Meier risk stratification for KIRC.

# H    CLINICAL UTILITY

## H.1    MULTIVARIABLE SURVIVAL ANALYSIS

To assess whether SlotSPE provides prognostic value beyond clinical variables (age, sex, stage, and neoadjuvant treatment), which are empirically shown as crucial prognostic factors (Xu et al., 2019; Tomasello et al., 2017; Edge & Compton, 2010; Yu et al., 2022), we performed multivariable Cox regression (Cox, 1972) using both clinical covariates and our model's predicted risk scores. For each cohort, we merged the risk prediction of SlotSPE with these clinical variables on a per-patient basis. Importantly, the risk scores were obtained from the validation folds, ensuring that the model had no access to these samples during training. We then fitted Cox models across the five validation folds to quantify the incremental prognostic contribution of SlotSPE. In Table 10, we report the time-dependent AUC (Heagerty et al., 2000; Heagerty & Zheng, 2005), integrated brier score (IBS) (Graf et al., 1999; Gerds & Schumacher, 2006), and C-index for two configurations: (i) *clinical-only*, using only the clinical covariates, and (ii) *clinical + model*, where the predicted risk score of SlotSPE is included as an additional covariate. We also report the corresponding improvement ΔC-index. Across all five cohorts, adding the risk score of SlotSPE to the clinical covariates consistently improves the C-index, with ΔC-index ranging from 0.023 (KIRC) to 0.054 (LUAD). Time-dependent AUC likewise increases or remains comparable when including the model, while IBS is generally stable or reduced (e.g., substantial IBS reduction for KIRC and LUSC). These results indicate that Slot-SPE captures complementary prognostic information that is not fully explained by routine clinical variables, showing the clinical utility potential to incorporate our model's prediction as a clinically useful prognostic factor when combined with established clinical factors.

## H.2    CALIBRATION, DECISION ANALYSIS, AND RISK STRATIFICATION

To further evaluate the clinical utility of SlotSPE, we examine calibration and decision curve analysis (DCA) on the KIRC cohort. As shown in Figure 11(a), the 60-month calibration curve demonstrates strong agreement between predicted and observed survival probabilities, indicating that SlotSPE provides well-calibrated risk estimates. The decision curve in Figure 11(b) shows that SlotSPE yields positive net benefit across a clinically meaningful range of threshold probabilities. Within this range, SlotSPE consistently yields higher net benefit than both the "treat-all" and "treat-none"

strategies, demonstrating potential practical value for supporting clinical decision-making. Finally, the KM analysis in Figure 11(c) shows a clear and statistically significant separation between high- and low-risk groups, confirming that SlotSPE offers meaningful prognostic stratification.

# I INTERPRETABILITY

## I.1 PATIENT-LEVEL INTERPRETABILITY

**Cross-modality Alignment.** We visualize slot assignment maps for both histological and genomic slots (Figure 12A). Slots from both modalities consistently segment similar cellular and tissue regions within the WSI, indicating that SlotSPE captures event-level alignment across modalities. At the same time, differences emerge: morphology-driven slots emphasize structural tissue patterns, whereas genomics-driven slots highlight regions reflecting underlying molecular features.

**Intra- and Inter-modal Interactions.** For the histological modality (Figure 12B, first row), we display the attention map for its assigned slot, and further show the top-5 patches with the highest attention. An experienced pathologist provided expert annotations for each of these patches. These attention maps provide insight into which tissue regions the model identifies as the most prognostically informative. For the pathway modality, we visualize the assignment maps of pathways for omics-derived slots and also report the top-3 most relevant and irrelevant pathways per slot, revealing how the model differentiates biologically meaningful signals (Figure 12C). Then, to analyze cross-modal interactions, we connect pathway-level attention scores with the corresponding attention map of genomic slot on WSIs (Figure 12B, second row). Another case in UCEC is shown in Figure 13

Overall, the interpretability analysis highlights the ability of the model to establish event-level alignment across modalities, as well as showing the potential to make novel hypotheses by identifying unexpected pathway–morphology correspondences for further biological validation.

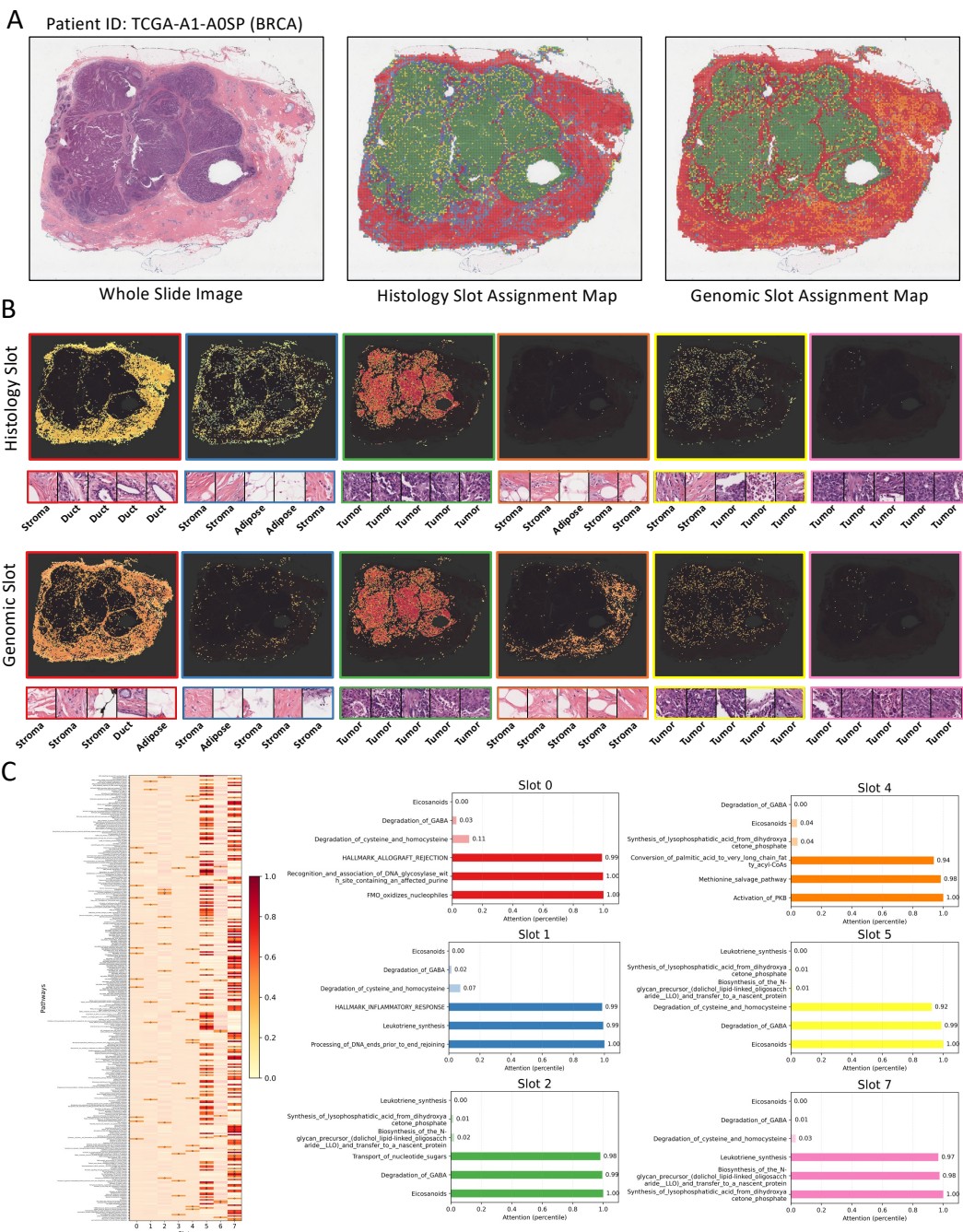

Figure 12: Case study of patient-level interpretability in BRCA. (A) From left to right: original WSI, assignment map of histology-derived slots, and assignment map of omics-derived slots. (B) Attention maps for each slot highlighting the top-5 most relevant patches. (C) Assignment maps of pathways for omics-derived slots with corresponding attentions, showing the top-3 relevant and irrelevant pathways per slot.

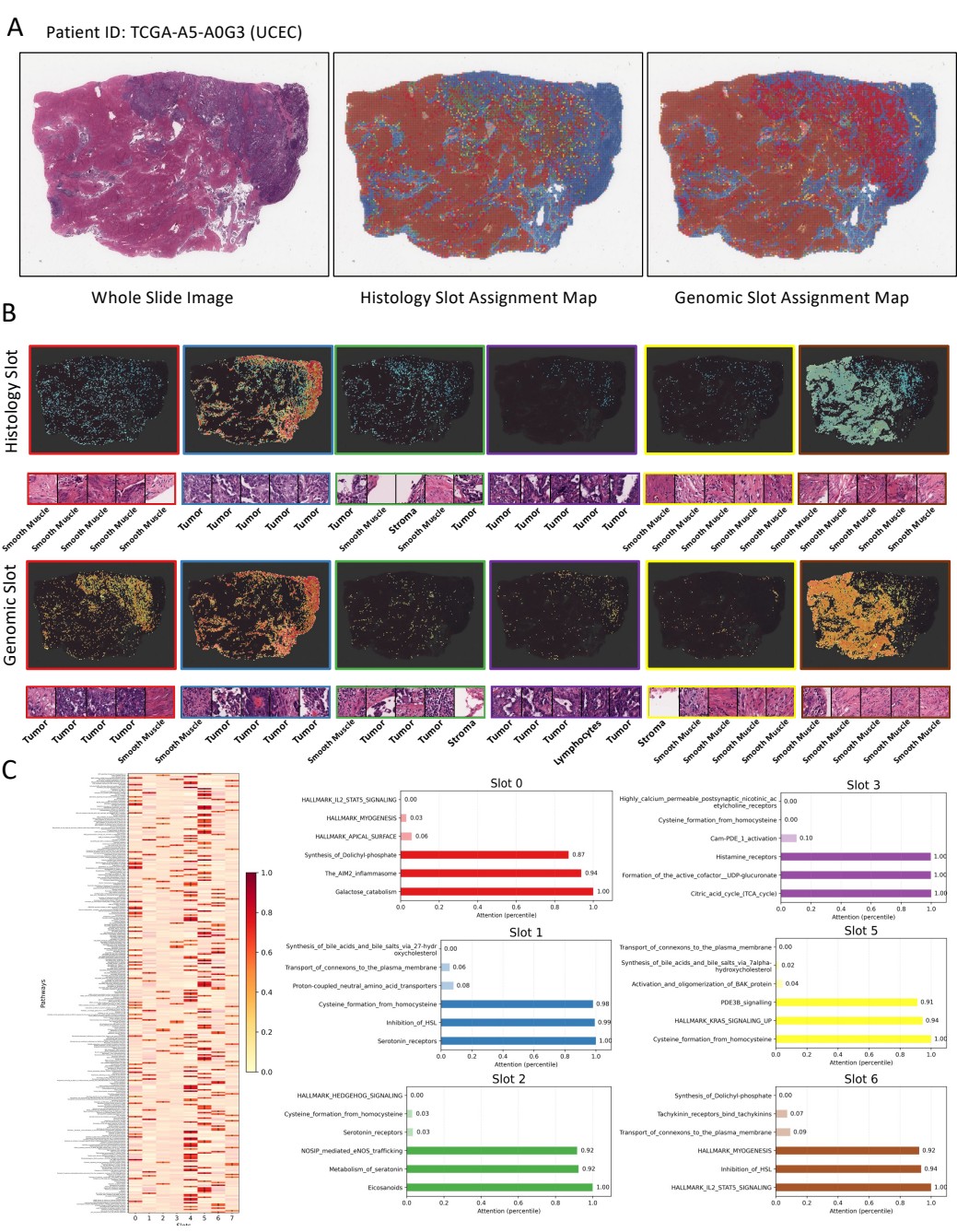

Figure 13: Case study of patient-level interpretability in UCEC. (A) From left to right: original WSI, assignment map of histology-derived slots, and assignment map of omics-derived slots. (B) Attention maps for each slot highlighting the top-5 most relevant patches. (C) Assignment maps of pathways for omics-derived slots with corresponding attentions, showing the top-3 relevant and irrelevant pathways per slot.

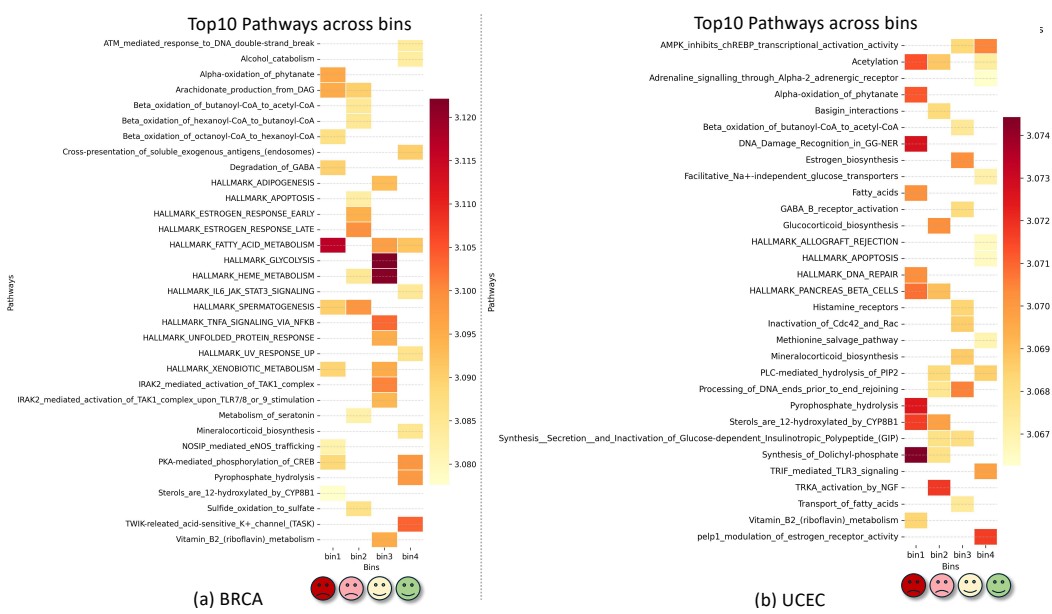

Figure 14: Cohort-level pathway analysis for BRCA and UCEC. Patients are stratified into four risk groups, and top-10 pathways are identified per group based on slot attentions. Distinct patterns emerge across bins, highlighting biologically meaningful differences between high- and low-risk groups.

## I.2 COHORT-LEVEL INTERPRETABILITY

To extend interpretability beyond individual patients, we analyze cohort-level pathway attention patterns in the genomic modality. For each patient, we identify the slot with the highest retention score from the MoE decoder and extract its pathway-level attentions. Patients are then stratified into four risk groups (bins) based on their survival time. Within each group, pathway attentions are averaged to obtain group-level profiles, and the top-10 pathways per bin are reported (Figure 14).

In BRCA, for example, high-risk patients (bin1) are enriched in fatty acid and xenobiotic metabolism pathways, reflecting the metabolic reprogramming characteristic of aggressive tumors (Pavlova & Thompson, 2016; Röhrig & Schulze, 2016). In contrast, low-risk patients (bin4) exhibit stronger DNA repair and immune-related pathways, consistent with improved prognosis (Nolan et al., 2017; Denkert et al., 2018). Such cohort-level analyses illustrate the potential of SlotSPE to uncover pathway–risk associations and generate hypotheses for further biological validation.

