# OpenReview forum: "Structural Prognostic Event Modeling for Multimodal Cancer Survival Analysis"
_ICLR.cc/2026/Conference — ICLR 2026 Poster_

### Official Review · Reviewer_HWAP · 2025-10-15

**Soundness:** 3
**Presentation:** 3
**Contribution:** 2
**Rating:** 6
**Confidence:** 3

**Summary:**

This paper introduces SlotSPE, a novel framework for multimodal cancer survival analysis that integrates histology images (WSIs) and genomic data. The core innovation is to model sparse, patient-specific "prognostic events" by compressing high-dimensional inputs into a compact set of dynamic "slots" using slot attention. The model selectively activates the most predictive slots for each patient via a Mixture-of-Experts (MoE) style decoder, enhancing personalization and sparsity. A key feature is a biologically-guided cross-modal reconstruction task, where the model learns to predict gene expression from histology images, thereby enforcing meaningful alignment and enabling robust performance even with missing genomic data. Extensive experiments on ten TCGA cancer cohorts demonstrate that SlotSPE significantly outperforms existing state-of-the-art methods in predictive accuracy, robustness, and interpretability.

**Strengths:**

1. Strong Empirical Evidence: The claims are well-supported by extensive experiments across ten TCGA cancer cohorts.
2. Enhanced Interpretability: The model provides a structured way to interpret its predictions.
3. This model can better capture the sparse and individualized nature of cancer drivers.
4. Biologically Guided Alignment: A key strength is the cross-modal reconstruction task, where omics-derived slots are used to predict gene expression from histology images.

**Weaknesses:**

1. The article does not provide a detailed analysis of training time or computational resource usage compared to other methods (e.g. flops), which makes it difficult to assess its practical scalability.
2. There are a few baselines for comparison under missing modalities. Although some baselines do not take into account the actual modality, you can manually modify the code for testing.
3. Hyperparameter Sensitivity: The model has numerous hyperparameters, including the number of slots for each modality, the number of selected top-K. Are there any results of ablation of these hyperparameters?

**Questions:**

1. Your method is somewhat similar to AdaMHF[1], both consider efficient and input sparse. Please explain the difference and compare and cite it.
[1] Zhang S, Lin X, Zhang R, et al. AdaMHF: Adaptive Multimodal Hierarchical Fusion for Survival Prediction[J]. arXiv preprint arXiv:2503.21124, 2025.
2. The article does not provide a detailed analysis of training time or computational resource usage compared to other methods (e.g. flops), which makes it difficult to assess its practical scalability.
3. There are a few baselines for comparison under missing modalities. Although some baselines do not take into account the actual modality, you can manually modify the code for testing.
4. Hyperparameter Sensitivity: The model has numerous hyperparameters, including the number of slots for each modality, the number of selected top-K. Are there any results of ablation of these hyperparameters?

---

> ### Author Response · Authors · 2025-11-21
> **Response to Reviewer HWAP [1/2]**
>
> We thank the reviewer for the constructive feedback and for recognizing our contributions. We sincerely appreciate your time to read the paper. In the following, your comments are first stated and then followed by our point-by-point responses. All updates made in response to the reviews are highlighted in **green** in the revised manuscript.
>
> ---
>
> > **Q1) Your method is somewhat similar to AdaMHF[1], both consider efficient and input sparse. Please explain the difference and compare and cite it. [1] Zhang S, Lin X, Zhang R, et al. AdaMHF: Adaptive Multimodal Hierarchical Fusion for Survival Prediction[J]. arXiv preprint arXiv:2503.21124, 2025.**
> >
>
> We thank the reviewer for raising this point. While both AdaMHF and SlotSPE consider efficient and input sparse, they address this challenge in different ways. AdaMHF performs adaptive token selection and aggregation to filter redundancy at the token level, whereas SlotSPE reframes survival prediction through structured prognostic event modeling, learning a small set of high-level, patient-specific “event slots” that capture interpretable prognostic factors. In addition, SlotSPE incorporates biologically guided cross-modal reconstruction, a mechanism that is absent in AdaMHF. We have added a citation to AdaMHF and included a discussion in the related works section **(Section 2, Page 3)**.
>
> Regarding empirical comparison, the official implementation of AdaMHF is currently unavailable (the link provided in the paper is invalid). We made our best effort to re-implement the method in the limited time and report the strongest performance we could obtain in the table below. However, in the absence of an official implementation, we acknowledge that such comparisons may not be fully fair, and therefore, we present these results here primarily for reference.
>
> | Methods | BRCA | COADREAD | KIRC | UCEC | LUAD | LUSC | HNSC | SKCM | BLCA | STAD | Overall |
> | --- | --- | --- | --- | --- | --- | --- | --- | --- | --- | --- | --- |
> | AdaMHF | 0.683 | 0.706 | 0.767 | 0.785 | 0.646 | 0.622 | 0.608 | 0.687 | 0.661 | 0.623 | 0.679 |
> | SlotSPE | 0.779 | 0.773 | 0.815 | 0.813 | 0.683 | 0.634 | 0.642 | 0.688 | 0.708 | 0.671 | 0.721 |
>
> ---
>
> > **W1&Q2) The article does not provide a detailed analysis of training time or computational resource usage compared to other methods (e.g. flops), which makes it difficult to assess its practical scalability.**
> >
>
> We appreciate the reviewer’s concern, which was also raised by Reviewer 7qhV, and we incorporated both sets of feedback. We would first like to clarify that our initial submission did not include empirical efficiency metrics—only the theoretical **big-O** analysis—because such practical efficiency comparison is difficult as the results can be highly sensitive to the choice of auxiliary architectural components and low-level engineering details.  Our primary objective was to reframe multimodal survival prediction through ***structural prognostic event modeling***, which inherently reduces the computational burden of modeling extensive intra- and inter-modal interactions. Nonetheless, in response to the reviewer’s request, we have added a comprehensive efficiency evaluation in **Section 4.6 (Page 9)** and **Appendix G (Page 30-31)**, including inference and training runtime, peak GPU memory, and a detailed module-level cost breakdown, all measured under a unified hardware configuration (NVIDIA RTX A6000, 48GB). These results show that SlotSPE achieves a strong performance–efficiency balance while maintaining a competitive computational footprint. Notably, during inference, SlotSPE requires **minimal memory** and achieves **the second-fastest runtime** among all compared methods.
>
> ---
>
> > **W2&Q3) There are a few baselines for comparison under missing modalities. Although some baselines do not take into account the actual modality, you can manually modify the code for testing.**
> >
>
> We thank the reviewer for this suggestion. While it is possible to manually modify baseline implementations to simulate missing modalities, most existing multimodal survival models are **not designed** to handle missing-modal scenarios, and forcing them into this setting may compromise the comparison fairness. Nevertheless, to provide additional evidence, we evaluated several baselines under missing-genomics conditions in **Appendix E (Page 23,25)**.
>
> As shown in the revised manuscript, **Table 6 (Page 25)**, even strong multimodal methods such as MOTCAT and CMTA experience a substantial performance drop when genomic features are absent, often falling below high-performing single-modal baselines like ABMIL. In contrast, SlotSPE remains robust owing to its explicit cross-modal reconstruction—originally introduced to facilitate better cross-modal alignment—which also provides clear benefits under missing-genomics conditions

---

> ### Author Response · Authors · 2025-11-21
> **Response to Reviewer HWAP [2/2]**
>
> > **W3&Q4) Hyperparameter Sensitivity: The model has numerous hyperparameters, including the number of slots for each modality, the number of selected top-K. Are there any results of ablation of these hyperparameters?**
> >
>
> We thank the reviewer for raising this question. Our original submission already included the ablation on the slot numbers and slot-attention iterations. Multiple reviewers expressed interest in broader hyperparameter sensitivity. In response, we have expanded the analysis in the revision. Specifically, **Appendix F.3–F.4 (Page 26-29)** now provides detailed ablations covering:
>
> - **F.3.1 – Slot numbers (Page 26-27)**
> - **F.3.2 – Slot-attention iterations (Page 28)**
> - **F.4.1 – Top-K slot selection (Page 28-29)**
> - **F.4.2 – Softmax temperature (Page 28-29)**
>
> These experiments show that SlotSPE is stable across a wide range of hyperparameters. Performance generally improves with **moderate slot numbers**, **more slot-attention iterations**, **moderate Top-K**, and **low softmax temperature**, and we do not observe drastic degradation under reasonable settings. Overall, the model demonstrates strong robustness to its key hyperparameters.

---

### Official Review · Reviewer_xj7Z · 2025-10-25

**Soundness:** 3
**Presentation:** 3
**Contribution:** 2
**Rating:** 6
**Confidence:** 2

**Summary:**

This paper focuses on the problem of modeling intra- and inter-modal interactions effectively and efficiently. The authors proposed SlotSPE, a Slot-based Structural Prognostic Event modeling method. SlotSPE compress each patient’s multimodal inputs into compact, modality-specific sets of mutually distinctive slots using slot attention. By using the slots representations as encoding s for prognostic events, the method enables both efficient and effective modeling of complex intra- and inter-modal interactions, while also facilitating seamless incorporation of biological priors that enhance prognostic relevance.

**Strengths:**

1. The evaluation experiments is comprehensive and convincing.
2. The paper is overall well-written and easy to follow.
3. The experimental results shows that the value of most evaluation metrics of the proposed method obviously outperform the baseline methods.
4. The proposed selective slot is simple but effective and novel.

**Weaknesses:**

1. The ablation study, though detailed, could be extended to include comparisons with more recent foundation models.

**Questions:**

1. How sensitive is SlotSPE to the number of slots and gating hyperparameters?
2. Have the authors examined whether the reconstructed genomic features can be used directly for downstream biological interpretation?

---

> ### Author Response · Authors · 2025-11-21
> **Response to Reviewer xj7Z**
>
> We thank the reviewer for the constructive feedback and for recognizing our contributions. We sincerely appreciate your time to read the paper. In the following, your comments are first stated and then followed by our point-by-point responses. All updates made in response to the reviews are highlighted in **green** in the revised manuscript.
>
> ---
>
> > **W1) The ablation study, though detailed, could be extended to include comparisons with more recent foundation models.**
> >
>
> We thank the reviewer for this valuable suggestion. In the revision, we have expanded our ablation study to include comparisons with recent pathology foundation models, in addition to the original ResNet50 experiment. As shown in **Figure 7 in Appendix F.2 (Page 26-27)**, we evaluate SlotSPE with CONCH[1], CONCH v1.5 (published in TITAN[2]), and UNI[3]. While all pathology-specific encoders generally improve performance relative to ImageNet pretraining, UNI yields the best results, and SlotSPE remains consistently strong across all encoder choices. These findings demonstrate that SlotSPE is (i) robust even under a weaker, non–domain-specific encoder, and (ii) fully compatible with recent pathology foundation models.
>
> **REFERENCES**
>
> [1]  Lu, Ming Y., et al. "A visual-language foundation model for computational pathology." *Nature medicine* 30.3 (2024): 863-874.
>
> [2] Ding, Tong, et al. "A multimodal whole-slide foundation model for pathology." *Nature Medicine* (2025): 1-13.
>
> [3] Chen, Richard J., et al. "Towards a general-purpose foundation model for computational pathology." *Nature medicine* 30.3 (2024): 850-862.
>
> ---
>
> > **Q1) How sensitive is SlotSPE to the number of slots and gating hyperparameters?**
> >
>
> We thank the reviewer for raising this question. We provide a detailed sensitivity analysis in **Appendix F.3.1 (Page 26-27)** and **Appendix F.4 (Page 28-29)** in the revised manuscript.
>
> For the number of slots, as shown in **Figure 8 (Page 27)**: performance generally improves when increasing slots from very small values, but quickly stabilizes once reaching a moderate range (16–32 slots). Beyond this point, additional slots offer little or no gain and may introduce redundancy. We also include experiments using only 8 slots per modality (shown in **Table 4 (Page 22)**), where SlotSPE still achieves a strong C-index (overall 0.706), exceeding the best baseline by **1.4%**, indicating that the slots effectively capture structured prognostic events through highly compressed representations.
>
> For gating hyperparameters, we analyze Top-K slot selection and softmax temperature in **Appendix F.4 (Page 28-29).** As shown in **Figure 9(b) (Page 28)**, very small K leads to under-activation and poor performance, while very large K forces non-discriminative slots to activate. Mid-range settings perform similarly, and we adopt 25% as patients are divided into four risk groups. **Figure 9(c)** **(Page 28)** shows that the softmax temperature is also stable within a reasonable range: high temperatures produce diffuse gating and degrade performance, whereas lower temperatures sharpen slot selection. The setting temp = 0.01 provides the best results across datasets.
>
> ---
>
> > **Q2) Have the authors examined whether the reconstructed genomic features can be used directly for downstream biological interpretation?**
> >
>
> We appreciate the reviewer’s insightful question. Examining whether the reconstructed genomic features can support downstream biological interpretation is indeed an interesting direction; however, rigorous validation requires a broad multidisciplinary effort that goes beyond the scope of this paper. In our work, the reconstructed omics representations are used to facilitate better cross-modal alignment and improve robustness under missing-modality scenarios, where they demonstrably enhance survival prediction performance. Nevertheless, we agree that leveraging these reconstructed features for biological interpretation is a promising avenue for future research.

---

### Official Review · Reviewer_7qhV · 2025-10-31

**Soundness:** 3
**Presentation:** 3
**Contribution:** 3
**Rating:** 6
**Confidence:** 1

**Summary:**

The paper proposes a slot-based, patient-specific representation for multimodal prognosis: whole-slide images and pathway-level transcriptomics are compressed into a few sparsely gated “prognostic event” slots; cross-modal reconstruction guided by biological priors aligns morphology and omics and enables imputation when omics is missing. Across multiple TCGA cohorts, the method shows consistent C-index gains and argues lower computational complexity by interacting at the slot level rather than instance level. While promising for interpretability and robustness, the work lacks empirical evidence for its efficiency claims (runtime/throughput/memory/scaling) and does not integrate clinical covariates or report multivariable analyses, calibration, or decision utility.

**Strengths:**

- Patient-specific slot representation with sparse gating focuses on a small set of salient prognostic factors, improving interpretability.

- Cross-modal reconstruction with pathway priors aligns WSI and transcriptomics and supports missing-omics scenarios.

- Consistent performance gains over baselines across multiple TCGA cohorts, with competitive single-modality ablations.

- Architecture and losses are clearly specified, facilitating ablations and reproduction.

**Weaknesses:**

- Efficiency evidence missing: No wall-clock time, throughput (slides/s), peak GPU memory, or scaling curves vs. #slots/Top-K/#patches under identical hardware/batch settings.

- Clinical variables omitted: No inclusion of age/stage/treatment etc., and no multivariable survival analysis, calibration, or decision-curve analysis to establish clinical utility.

**Questions:**

1. Efficiency & scaling
  - Please report training/inference time, throughput, and peak GPU memory for your method and strong baselines on the same hardware and batch size.
  - Provide scaling curves for time/memory as functions of #slots (S), Top-K, and #WSI patches, and discuss speed/accuracy and memory/accuracy trade-offs.
  - Break down module-level costs (WSI encoder, omics encoder, slot interactions, reconstruction) to identify bottlenecks.

2. Clinical utility
  - Incorporate clinical covariates (age, sex, stage, treatment, etc.) and report multivariable Cox/discrete-time results for: (i) clinical-only, (ii) model-only, and (iii) clinical + model, with ΔC-index, time-dependent AUC, and IBS.
  - Add calibration plots and decision curve analysis (DCA), and propose a concrete high/medium/low-risk stratification to illustrate potential clinical use.

---

> ### Author Response · Authors · 2025-11-21
> **Response to Reviewer 7qhV [1/2]**
>
> We thank the reviewer for the constructive feedback and for recognizing our contributions. We sincerely appreciate your time to read the paper. In the following, your comments are first stated and then followed by our point-by-point responses. Updates made in response to the reviews are highlighted in **green** in the revised manuscript.
>
> ---
>
> > **W1&Q1) Efficiency & Scaling: No wall-clock time, throughput (slides/s), peak GPU memory, or scaling curves vs. #slots/Top-K/#patches under identical hardware/batch settings.**
> >
>
> We appreciate this important point. We would first like to clarify that we did not initially include these measurements because practical efficiency comparison is difficult, as the results can be highly sensitive to the choice of auxiliary architectural components and low-level engineering details. Our primary objective was to reframe multimodal survival prediction through ***structural prognostic event modeling***, which inherently reduces the computational burden of modeling extensive intra- and inter-modal interactions. This stands in contrast to existing baselines that rely on heuristics such as skipping self-attention, and is reflected in our substantial performance gains. Nonetheless, in response to the reviewer’s request, we have conducted a comprehensive efficiency evaluation and added the analysis in **Section 4.6 (Page 9)** and **Appendix G (Page 30-31)** of the revised manuscript.
>
> > Q1.1 Please report training/inference time, throughput, and peak GPU memory for your method and strong baselines on the same hardware and batch size.
> >
>
> In the revision, we compare SlotSPE with the four strongest multimodal baselines from Table 1 under identical hardware (an NVIDIA RTX A6000, 48 GB) and batch (batch size = 1, as several official implementations do not support batch training) settings. The inference and training runtime and peak GPU memory usage are reported in **Figure 4 in**  **Section 4.6 (Page 9)** and **Figure 10(a)(b) in Appendix G.1 and G.2 (Page 30-31).** Given that the batch size is 1, throughput can be directly computed as 1 / runtime per batch. Although our work does not primarily focus on model lightweighting or speed optimization, SlotSPE still demonstrates an excellent balance between efficiency and performance compared with other methods—achieving the highest C-index while maintaining low memory consumption and runtime. Notably, during inference, SlotSPE exhibits **minimal memory** consumption and **the second-fastest runtime** among all compared methods.
>
> > Q1.2 Provide scaling curves for time/memory as functions of #slots (S), Top-K, and #WSI patches, and discuss speed/accuracy and memory/accuracy trade-offs.
> >
>
> We would first like to clarify that the Top-K selection does not affect runtime or memory, because it is implemented as a binary mask applied on top of all slots; i.e., varying K only affects the performance. The number of WSI patches is fixed across all compared methods (4,096 randomly sampled patches during training and all patches during inference), so varying the patch count was not included in this analysis.
>
> Accordingly, we report runtime and memory as a function of the number of slots, summarized in the table below. In this experiment, we follow the same batch size and patch count configuration as described in Appendix C: batch size = 32 with 4,096 sampled patches for training, and batch size = 1 with all patches for inference. We set the number of slots to be the same for both modalities and vary it jointly from 8 to 128.
>
> As shown in the results, increasing the slot count primarily impacts **batch training**, where increasing slots from 8 to 128 (a 16× increase) leads to only a ~2.5× increase in runtime and memory. In contrast, **inference** is much less sensitive: runtime increases by only ~1.1× and memory by ~1.7×. As we discussed in the slot numbers ablations in **Appendix F.3.1 (Page 26-27)**, performance improves as the slot count increases but quickly saturates once reaching a moderate number (16–32 slots). Beyond this range, additional slots provide little or no benefit. Therefore, to balance speed vs. performance and memory vs. performance, using a moderate number of slots (16–32 slots) is sufficient—it delivers strong performance without unnecessary computational overhead.
>
> | Training |  |  |
> | --- | --- | --- |
> | Slot Number | Time (s)/grad_step | Peak Memory (MB) |
> | 8 | 0.205 | 7991.38 |
> | 16 | 0.224 | 8765.66 |
> | 32 | 0.267 | 10311.14 |
> | 64 | 0.333 | 13433.33 |
> | 128 | 0.482 | 19797.58 |
> | Inference  |  |  |
> | slot_num | Time (s)/sample | Peak Memory (MB) |
> | 8 | 0.036 | 637.70 |
> | 16 | 0.037 | 637.86 |
> | 32 | 0.040 | 700.79 |
> | 64 | 0.038 | 826.65 |
> | 128 | 0.040 | 1078.90 |

---

> ### Author Response · Authors · 2025-11-21
> **Response to Reviewer 7qhV [2/2]**
>
> > Q1.3 Break down module-level costs (WSI encoder, omics encoder, slot interactions, reconstruction) to identify bottlenecks.
> >
>
> We appreciate the reviewer’s request. A module-level breakdown has been added to **Appendix F.3.1** **(Figure 10(c), Page 30-31)**. As shown there, the reconstruction branch is the major bottleneck, while slot attention and all other modules contribute comparatively minor overhead. These reconstruction components are used only during training and are disabled at inference unless the omics modality is missing, in which case the cross-modal reconstruction head is activated.
>
> ---
>
> > **W2&Q2) Clinical variables omitted: No inclusion of age/stage/treatment etc., and no multivariable survival analysis, calibration, or decision-curve analysis to establish clinical utility.**
> >
>
> We fully agree that clinical utility is important. However, establishing *validated* clinical utility requires extensive multidisciplinary efforts—spanning pathology, oncology, biostatistics, and prospective evaluation—which goes beyond the scope of this paper but represents a valuable direction for future exploration. Our original submission, therefore, provides risk stratification and interpretability as initial evidence of potential clinical application. In response to the reviewer’s suggestion, we have now included a clinical-utility analysis in **Appendix H (Page 32)**, which further demonstrates the model’s potential applicability in clinical settings.
>
> > Q2.1 Incorporate clinical covariates (age, sex, stage, treatment, etc.) and report multivariable Cox/discrete-time results for: (i) clinical-only, (ii) model-only, and (iii) clinical + model, with ΔC-index, time-dependent AUC, and IBS.
> >
>
> To address the reviewer’s question, we have added a clinical-utility analysis in **Appendix H.1 (Table 10, Page 32)**. In this analysis, we evaluate whether SlotSPE provides prognostic value beyond established clinical covariates (age, sex, stage, treatment) using multivariable Cox regression. We compare (i) **clinical-only** versus (ii) **clinical + model** configurations (we do not repeat the model-only results here, as their standalone performance is already reported in Table 1). As shown in **Table 10 (Page 32),** across five cohorts, adding SlotSPE’s predicted risk score to clinical covariates consistently improves the C-index (ΔC-index: 0.023–0.054), while time-dependent AUC increases or remains comparable. IBS is also stable or reduced. The results demonstrate that SlotSPE provides complementary prognostic information beyond routine clinical factors.
>
> > Q2.2 Add calibration plots and decision curve analysis (DCA), and propose a concrete high/medium/low-risk stratification to illustrate potential clinical use.
> >
>
> We thank the reviewer for this suggestion. In the revised manuscript, we added calibration plots, decision curve analysis, and a concrete risk-stratification evaluation, as presented in **Appendix H.2 (Figure 11)**. The results show that SlotSPE is well-calibrated at 60 months, provides clear positive net benefit across clinically meaningful threshold ranges in DCA, and yields statistically significant separation between high- and low-risk groups in KM analysis (consistent with Section 4.4). These findings further support the model’s potential clinical use.

---

### Official Review · Reviewer_kV7i · 2025-10-31

**Soundness:** 3
**Presentation:** 1
**Contribution:** 2
**Rating:** 2
**Confidence:** 5

**Summary:**

This paper proposes a structural prognostic event modeling framework for multimodal survival prediction. The proposed method identifies the patient-specific prognostic events through slot-based representation learning, integrates a cross-modal reconstruction mechanism embedded with biological priors to enhance modal alignment and robustness, and delivers superior and stable performance on ten cancer cohorts. Moreover, the framework demonstrates improved interpretability through the structured decoupling ability of the proposed representation.

**Strengths:**

1. This paper adopts a slot-attention module to survival analysis task and further introduces a Mixture-of-Experts (MoE) mechanism for selective slot activation.


2. Incorporating reconstruction regularizations preserves the information in input features and enhances the model’s robustness under missing-modality scenarios.


3. The proposed framework exhibits the generalization across ten cancer survival datasets, offers good interpretability.

**Weaknesses:**

1. The novelty of this work is limited, as both the Slot Attention and MoE modules have been widely used across various tasks, including survival analysis.

2. The overall writing and structural organization of the paper are quite unclear:

    (1) In the Selective Slot Activation section, the text notes “Conceptually, each slot is treated... A lightweight gating function $\phi$ predicts a retention score for each slot.” The description claims that the output is of dimension $N_t$, while according to the accompanying formula, it should be a scalar score—which is inconsistent.

    (2) In the paragraph “Then the top-K slots are selected using the Gumbel-Top-K...”, the variable $\widetilde{w}$ is of dimension $S$, where $S$ denotes the number of slots before selection. It is unclear whether the softmax is applied to the selected slots or to all slots.

    (3) Throughout the paper, many variables—especially those with subscripts—lack clear definitions, which seriously hinders readability.

    (4) Figure 2 is not introduced or referenced in the main text, making its purpose and relevance unclear.

    (5) In the paragraph “Given the omics-derived slots $S_g$,...”, the specific role of $S_g$ is not clarified—does it serve as initialization weights or as fixed guidance for new slot generation?

    (6) Section 3.2 as a whole lacks appropriate references to figures, leading to poor readability.

    (7) The paper’s organization is confusing; for instance, to understand the “reconstruction” part, the reader must jump to Appendix B.3, which in turn refers to Equation (5) that appears in the following section—this disrupts the logical reading flow.

    (8) In Section 3.2 (Slots Interactions), the description of self-interaction and cross-attention mechanisms is vague—it is not specified whether these operations are performed on all slots or only on the selected ones.

    (9) In Section 3.3 (Training and Inference), the reader is directed to Appendix B.5 for the total loss, yet the survival loss is not clearly defined. What exactly constitutes the “standard survival prediction loss”? Moreover, without Equation (18), it is impossible to infer the last two terms of $L_{\mathrm{surv}}$ from the main text, which indicates significant ambiguity in the writing.

3. The authors are encouraged to clarify the unique aspects and improvements of the MoE mechanism used in this work in comparison to the previously published paper such as [1], in order to emphasize the genuine novelty of the proposed approach.

    [1] From Single-Cancer to Pan-Cancer Prognosis: A Multi-Modal Deep Learning Framework for Survival Analysis with Robust Generalization Capability.

**Questions:**

Please refer to the **Weaknesses**.

---

> ### Author Response · Authors · 2025-11-21
> **Response to Reviewer kV7i [1/2]**
>
> We thank the reviewer for your time to read the paper. In the following, your comments are first stated and then followed by our point-by-point responses. Updates made in response to the reviews are highlighted in **green** in the revised manuscript.
>
> ---
>
> > **W1. The novelty of this work is limited, as both the Slot Attention and MoE modules have been widely used across various tasks, including survival analysis.**
>
> We first clarify that our contribution is NOT intended to be framed as “the first to apply method/building block ABC (e.g., Slot Attention, Transformers, deep networks) to domain XYZ (e.g., survival analysis).” Such a framing neither guarantees meaningful scientific progress nor leads to productive discussion in applied research. While we could state---accurately, to the best of our knowledge---that we are the first to apply Slot Attention to multimodal survival analysis, this is not the essence of our novelty. As outlined in the submission, our core contribution lies in reformulating the problem and translating it into a structural representation learning framework that enables novel and more effective solutions---of which our proposed method is one instantiation. Specifically, we reinterpret multimodal survival prediction as structural prognostic event modeling, and further as multimodal structural representation learning, leading to a fully learnable, patient-adaptive system. The resulting learning framework is mechanically distinct from prior methods and delivers substantially stronger empirical performance across diverse cohorts.
>
> ---
>
> > **W2. The overall writing and structural organization of the paper are quite unclear.**
>
> We thank the reviewer for the nine writing-related suggestions. We believe that three of these points arise from misunderstandings or oversights of our technical details, while the remaining comments---some of which overlap---pertain to writing clarity and can be addressed easily. We have incorporated the corresponding revisions into the updated manuscript.
>
> >(3)(4)(6)(7) Improving clarity and readability
>
> We have carefully reviewed the entire paper and revised the writing to improve clarity. All updates are highlighted in green in the revised manuscript. These revisions also incorporate relevant suggestions from the other reviewers.
>
> > (1) In the Selective Slot Activation section, the text notes “Conceptually, each slot is treated... A lightweight gating function $\phi$ predicts a retention score for each slot.” The description claims that the output is of dimension $N_t$, while according to the accompanying formula, it should be a scalar score—which is inconsistent.
>
> We believe the reviewer’s comment may stem from overlooking the definitions of $\boldsymbol{\ell}_k$ and $r_k$, which are clearly stated in the submission. Each slot (acting as an expert) produces its own class-logit prediction $\boldsymbol{\ell}_k \in \mathbb{R}^{N_t}$, whereas the referenced formula for the retention score $r_k$ represents the original routing weights (before top-K selection and softmax), so they are two different objects.
>
> > (2) In the paragraph “Then the top-K slots are selected using the Gumbel-Top-K...”, the variable $\tilde{w}_k$ is of dimension $S$, where $S$ denotes the number of slots before selection. It is unclear whether the softmax is applied to the selected slots or to all slots.
>
> We believe the reviewer’s comment may stem from overlooking the definitions of the $K$-hot mask variable $G$, which we introduced immediately above the referenced equation. The $G$ variable is a mask that "selects" and renormalize top-K "softmax-ed" slots---exactly shown in our Eq.(3).
>
>
> > (5) In the paragraph “Given the omics-derived slots $\mathbf{S}_g$, ...”, the specific role of $\mathbf{S}_g$ is not clarified—does it serve as initialization weights or as fixed guidance for new slot generation?
>
>
> The omics-derived slots $\mathbf{S}_g$ are learnable initializations, and are obtained from end-to-end learning from the omics (genomic) data. Note that, with such learnable initializations $\mathbf{S}_g$, our method is able to handle scenarios where input omics modality is missing. We have revised this description in the updated paper for improved clarity.

---

> ### Author Response · Authors · 2025-11-21
> **Response to Reviewer kV7i [2/2]**
>
> > (8) In Section 3.2 (Slots Interactions), the description of self-interaction and cross-attention mechanisms is vague—it is not specified whether these operations are performed on all slots or only on the selected ones.
>
> For self-interactions, we apply masked self-attention using only the top-K most self-predictive slots; however, through the residual connection, information from the remaining slots can still contribute indirectly. For cross-interactions, we use all slots without masking to fully capture inter-modal synergies, including those arising from slots that are originally less self-predictive within their own modality. We have revised the respective text for improved clarity.
>
> > (9) In Section 3.3 (Training and Inference), the reader is directed to Appendix B.5 for the total loss, yet the survival loss is not clearly defined. What exactly constitutes the “standard survival prediction loss”? Moreover, without Equation (18), it is impossible to infer the last two terms of from the main text, which indicates significant ambiguity in the writing.
>
> We defined the survival loss in **line 132 (Section 3) of the initial submission**, with a detailed explanation provided in Appendix B.1 (see Eq. (8)), following common practice in the survival analysis literature. However, for better clarity, we have updated the corresponding text and description. We also recognize that placing the full derivation in the appendix may be inconvenient for readers less familiar with the background, but including it in the main text is not feasible given the strict page limits of a conference submission.
>
> ---
>
> > **W3. The authors are encouraged to clarify the unique aspects and improvements of the MoE mechanism used in this work in comparison to the previously published paper such as [1], in order to emphasize the genuine novelty of the proposed approach.**
> >
>
> First, we note that the paper recommended by the reviewer appears to have limited relevance to our work, and we discovered its several claims and modelling choices lack clear justification. That said, we do not wish to initiate an unproductive debate about the soundness of another paper. Instead, we focus on clarifying the conceptual distinctions between the two approaches.
>
> The MoE mechanisms used in our framework and in the recommended paper are not comparable, as they serve fundamentally different modeling purposes. In our work, each expert corresponds to a distinct interpretable prognostic event, and the routing scores quantify the patient-specific manifestation of these events. This is central to our problem formulation and contributes directly to interpretability and predictive relevance.
>
> In contrast, the recommended paper uses MoE as a generic feature extractor without a clear modeling motivation, simply noting that MoE is used as a generic feature fuser, with claims such as "is capable of identifying common characteristics of different cancer types while adaptively extracting the features specific to each cancer type", without specifying what each expert represents nor why such decomposition is meaningful.
>
> Thus, the only genuine similarity is the superficial one: both architectures use routing-weighted combinations of expert outputs---i.e., both fall under the broad umbrella of MoE. Beyond this formal similarity, the modeling intent, interpretability, and functional role of the experts differ completely.

---

### Author Response · Authors · 2025-12-02
**Summary of Responses and Revisions**

Dear Area Chairs and Reviewers,

We sincerely thank the ACs and reviewers for your time and thoughtful evaluations of our submission. Below, we provide a concise summary of the key issues raised and our corresponding responses, to offer a clearer view of our contributions and the meticulous efforts invested in refining the manuscript to address all concerns.

**The majority of reviewers provided positive evaluations**, with three reviewers giving supportive scores for acceptance and highlighting the significance of our contributions. Reviewers 7qhV, xj7Z, and HWAP explicitly acknowledged the novelty of reframing multimodal survival prediction as structural prognostic event modeling, and further as multimodal structural representation learning, enabling a fully learnable and patient-adaptive system.

All reviewers acknowledged the breadth of our experimental evaluation—including comprehensive comparisons and ablation studies—and noted that the results are strong and impressive. To further strengthen the paper, reviewers suggested adding additional evidence on efficiency and clinical relevance. We conducted these analyses and found that our method **maintains superior prognostic performance while achieving substantially strong computational efficiency and offering interpretable clinical value**. These new results, now included in the revised submission, further reinforce the empirical strength of the work. We appreciate these constructive suggestions, which have meaningfully improved the paper.

We address reviewer kV7i's concerns separately, as the reviewer's assessment of novelty appears to stem from several misunderstandings. We respectfully note that novelty should be judged by methodological and conceptual contributions rather than keyword matching (e.g., "the first to apply method ABC to domain XYZ"). **Although we are, to the best of our knowledge, the first to apply Slot Attention to multimodal cancer survival analysis, this is *not* the novelty we emphasize, as simply being “the first to apply a method” does not necessarily constitute meaningful scientific advancement.** For the writing-clarity questions, we thoroughly reviewed the manuscript and refined the exposition. Most issues are minor and easily addressed, and others arise from misinterpretations of our technical definitions, which we clarify point-by-point. Notably, the other reviewers explicitly described the paper as **"well-written and easy to follow"** and stated that the "architecture and losses are clearly specified." Thus, we believe the revisions and clarifications provided fully address the concerns.

In conclusion, we are grateful for the reviewers' constructive feedback and insightful suggestions. We have invested our best efforts in addressing these questions and believe that our responses have effectively addressed all the concerns raised.

Best regards,\
The authors

---

### Meta-Review · Area_Chair_SbBR · 2026-01-02

**Summary:**

This paper proposes SlotSPE, a slot-based framework that reframes multimodal cancer survival prediction as structural prognostic event modeling, compressing WSIs and transcriptomics into sparse, patient-specific event representations with biologically guided cross-modal reconstruction. The problem is important, and most reviewers found the formulation novel at the modeling level, with strong and consistent empirical gains across ten TCGA cohorts, improved robustness under missing genomics, and enhanced interpretability.

The main concerns raised during review centered on (i) novelty perception (use of slot attention/MoE), (ii) clarity and organization in parts of the presentation, and (iii) practical efficiency and clinical relevance (runtime, scaling, and use of clinical covariates). The rebuttal and revision addressed these points: the authors clarified the conceptual novelty beyond component reuse, improved exposition, added comprehensive efficiency analyses (runtime, memory, scaling, and module-level costs), expanded foundation-model ablations, provided hyperparameter sensitivity studies, and included multivariable clinical analyses, calibration, and decision-curve evaluations. These additions materially strengthen confidence in both the technical contribution and practical relevance.  One reviewer remained negative, largely due to misunderstandings about definitions and novelty; however, the majority of reviewers were positive or borderline-positive, and the rebuttal resolves the substantive technical concerns. Remaining limitations (e.g., absence of prospective validation) are acknowledged and appropriate for a methods paper.

Overall, I recommend accepting of this paper.

**Reviewer Concerns:**

The rebuttal strengthened the paper and addressed nearly all substantive reviewer concerns. In particular, the authors added comprehensive efficiency and scalability evaluations (runtime, memory, and scaling with slot number), clinical utility analyses (multivariable Cox models, calibration, decision curve analysis), and extended ablations (slot number, Top-K, temperature, and encoder choice including recent pathology foundation models). Clarifications on novelty, MoE design intent, and slot interactions were technically sound and resolved several misunderstandings raised by one reviewer. Remaining limitations—such as the absence of prospective clinical validation—are appropriately acknowledged and do not affect the methodological contribution.

**Reviewer Scores:**

Reviewer kV7i (original: 2, reject): Likely increase to 4 or 6, as most cited issues (clarity, formulation consistency, novelty framing) were directly addressed, though skepticism about contribution may remain.

Reviewer 7qhV (original: 6): Likely unchanged, with efficiency and clinical analyses fully addressed.

Reviewer xj7Z (original: 6): Likely unchanged, as additional foundation-model experiments and sensitivity analyses resolved the main requests.

Reviewer HWAP (original: 6): Likely unchanged, given added efficiency analysis, missing-modality experiments, and hyperparameter ablations.

Overall, the rebuttal would plausibly shift the balance toward a consensus in favor of acceptance.

---

### Decision · Program_Chairs · 2026-01-26

Accept (Poster)